



# Isotopic constraints on wildfire derived HONO

Jiajue Chai[1,2], Jack E. Dibb[3], Bruce E. Anderson[4], Claire Bekker[1,2], Danielle E. Blum[1,5], Eric Heim[3], Carolyn E. Jordan[4,6], Emily E. Joyce[1,2], Jackson H. Kaspari[7], Hannah Munro[3], Wendell W. Walters[1,2], and Meredith G. Hastings[1,2]

[1] Institute at Brown for Environment and Society, Brown University, Providence, RI

[2] Department of Earth, Environmental and Planetary Sciences, Brown University, Providence, RI

[3] Institute for the Study of Earth, Ocean and Space, University of New Hampshire, Durham, NH

[4] NASA Langley Research Center, Hampton, VA

[5] Department of Chemistry, Brown University, Providence, RI

[6] National Institute of Aerospace, Hampton, VA

[7] Department of Chemistry, University of New Hampshire, Durham, NH

*Correspondence to*: Jiajue Chai (jiajue_chai@brown.edu)

**Abstract.** Nitrous acid (HONO) is an important precursor to hydroxyl radical (OH) that determines atmospheric oxidative capacity and thus impacts climate and air quality. Wildfire is not only a major direct source of HONO, but it also results in highly polluted conditions that favour heterogeneous formation of HONO from nitrogen oxides ($NO_x = NO + NO_2$) and nitrate on both ground and particle surfaces. However, these processes remain poorly constrained. To quantitatively constrain the HONO budget under various fire/smoke conditions, we combine a unique dataset of field concentrations and isotopic ratios ($^{15}N/^{14}N$ and $^{18}O/^{16}O$) of $NO_x$ and HONO, with an isotopic box model. Here we report the first isotopic evidence of secondary HONO production in near-ground wildfire plumes, and the subsequent quantification of relative importance of each pathway to total HONO production. Most importantly, our results reveal that nitrate photolysis plays a minor role (<5%) in HONO formation in daytime aged smoke, while photo-enhanced $NO_2$-to-HONO heterogeneous conversion contributes 85-95% to total HONO production, followed by OH+NO (5-15%). In nighttime, heterogeneous reduction of $NO_2$ catalysed by redox active species (e.g., iron oxide and/or quinone) is essential (≥75%) for HONO production in addition to surface $NO_2$ hydrolysis. Additionally, the $^{18}O/^{16}O$ of HONO is used for the first time to constrain the NO-to-$NO_2$ oxidation branching ratio between ozone and peroxy radicals. Our approach provides a new and critical way to mechanistically constrain atmospheric chemistry/air quality models.

## 1 Introduction

Vastly increased wildfire activity and intensity is a challenging issue in many parts of the world including the western United States, and it is strongly linked to warming surface temperatures and earlier spring snowmelt (Westerling, 2016). Wildfire is



a significant source of nitrogen oxides ($NO_x = NO + NO_2$) and nitrous acid (HONO), as well as other important trace gases and particulate matter. $NO_x$ serves as a key precursor to atmospheric ozone ($O_3$) and secondary aerosols in the presence of organic compounds; in wildfire plumes $NO_x$ can be a limiting factor to $O_3$ production owing to high emission molar ratios of non-methane organic carbon (NMOC) to $NO_x$ (Akagi et al., 2011; Jaffe and Briggs, 2012). HONO is a major daytime photolytic precursor of hydroxyl radical (OH) via R1 (Fig. 1) that determines the atmospheric oxidative capacity and

therefore the lifetimes of many other species in the atmosphere. Wildfire emitted HONO supplies the majority of OH in the first few hours after smoke emission in the daytime, and it greatly counteracts reduced OH production from $O_3$ photolysis caused by high particle loading reducing actinic flux (Jaffe and Briggs, 2012; Peng et al., 2020; Theys et al., 2020). Wildfire emitted $NO_x$ and HONO not only greatly impact the atmospheric chemistry in local regions close to the fire, but also contribute significantly to the reactive nitrogen (RN) burden thousands of kilometres downwind via transport and RN

cycling, especially when mixed with fossil fuel combustion emissions (Jaffe et al., 2013; McClure and Jaffe, 2018; Westerling et al., 2006; Westerling, 2016).

Despite their strong implication for air quality, climate, and human and ecosystem health, the budgets of wildfire-derived $NO_x$ and HONO are poorly constrained due to limited field measurements, high reactivity, and large spatiotemporal heterogeneity. Bottom-up approaches rely on limited emission factor measurements with uncertainty in HONO sources and chemistry; top-down approaches (i.e., satellite observations) have limited sensitivity in the lower troposphere and boundary

layer, and again, are limited by large uncertainties in HONO sources and chemistry to interpret the satellite measurements. Although gas phase reaction between OH and NO (R2 in Fig. 1) ubiquitously produces HONO, it is far from sufficient to explain the observed HONO levels in numerous studies given the fast photolysis during the day (Su et al., 2011). HONO, along with $NO_x$, can be directly emitted from various sources including vehicle exhaust, biomass burning (BB) and

microbially-driven soil emissions. In addition, it has been proposed that HONO can be produced from other RN species (e.g., $NO_2$ and nitrate) via various heterogeneous pathways (Fig. 1). Major secondary HONO production pathways during the day include heterogeneous $NO_2$ conversion on photoactive surfaces (R3) (Ammann et al., 1998; George et al., 2005; Stemmler et al., 2006), and heterogeneous photolysis of nitrate including particulate nitrate ($p\text{-}NO_3^-$) and nitric acid ($HNO_3$) via R4 (Ye et al., 2016; Zhou et al., 2011). In past studies, heterogeneous conversion of $NO_2$ to HONO on photoactive surfaces such as

organic surfaces (R3) has been proposed to explain a missing HONO source (Ammann et al., 1998; George et al., 2005; Stemmler et al., 2006; Wong et al., 2012). Organic surfaces exist in both aerosol particles and soils at the surface (e.g., humic acids) but there is major uncertainty associated with quantifying available surface area and the $NO_2$ uptake coefficient. During the night, surface (soils and aerosols) uptake is the predominant sink for HONO (R5), and heterogeneous conversion of $NO_2$ to HONO has been widely accepted as the major secondary HONO production source during the night (R6, R7).

Although it is clear that heterogeneous $NO_2$ hydrolysis (R6) can be a major pathway for nighttime HONO production (Finlayson-Pitts et al., 2003), recent work has also shown compelling evidence of faster HONO formation via reduction of $NO_2$ on inorganic surfaces (e.g., iron-bearing minerals) and organic surfaces (e.g., quinone-rich humic acid) in soils and



particulate matter (R7) (Scharko et al., 2017; Kebede et al., 2016; Martins-Costa et al., 2020). While the emission sources and heterogeneous pathways were hypothetically used to account for missing HONO sources (Stemmler et al., 2006; Su et al., 2011; Ye et al., 2016; VandenBoer et al., 2014; Donaldson et al., 2014a; Kebede et al., 2016; Scharko et al., 2017), their relative importance is poorly quantified due to large uncertainties associated with emission heterogeneity, surface area and composition, environmental condition (day versus night, temperature, relative humidity), quantification of heterogeneous reaction rate and knowledge gaps in detailed mechanisms. As a result, the HONO budget in wildfire smoke plumes remains poorly constrained.

Stable isotopes hold unique promise to provide rigorous constraints on sources, chemical processing pathways and sinks of reactive nitrogen species, as they reflect isotopic signatures associated with these processes. $\delta^{15}N$ (=$[(^{15}N/^{14}N)_{sample}/(^{15}N/^{14}N)_{air-N2}-1]\times1000‰$) has shown great potential to trace atmospheric origin of $NO_x$ and its fate as nitrate (Hastings et al., 2009), whereas $\delta^{18}O$ (=$[(^{18}O/^{16}O)_{sample}/(^{18}O/^{16}O)_{VSMOW}-1]\times1000‰$; VSMOW is Vienna Standard Mean Ocean Water) serves as a sensitive indicator for relative importance of major oxidants (i.e. $O_3$, $RO_2$ and OH) that lead to $NO_x$ conversion (Thiemens, 2006). In particular, $O_3$ has an exclusively high $\delta^{18}O$ as a result of mass-independent fractionation associated with its formation in the atmosphere and this anomaly is transferred to oxidized products such as $NO_2$, HONO and $HNO_3$ (Thiemens, 2006). By definition, $\varepsilon=(\alpha-1)\times1000‰$, with fractionation factor $\alpha$ referring to the rate coefficient ratio between the heavy isotopologue and the light isotopologue.

Using our recently developed and validated sampling techniques in combination with offline isotopic composition analyses (Chai et al., 2019), we characterized for the first time $\delta^{15}N$ of $NO_x$ and HONO as well as $\delta^{18}O$-HONO in ground-level wildfire plumes in the western US as part of two major field campaigns: Western Wildfire Experiment for Cloud Chemistry, Aerosol Absorption and Nitrogen (WE-CAN) in summer 2018, and Fire Influence on Regional to Global Environments Experiment – Air Quality (FIREX-AQ) in summer 2019. Here we report our findings based on samples collected in a mobile laboratory platform from three different wildfires; Rabbit Foot Fire (RF) in eastern Idaho, Williams Flats Fire (WF) in central Washington and Nethker Fire (NF) in northern Idaho (Fig. S1 and Fig. S2 in the supplement). Surface-based mobile sampling allowed us to characterize young nighttime (YN), young daytime (YD), mixed daytime (MD), aged nighttime (AN), and aged daytime (AD) smoke. Physical smoke age determination using meteorological parameters near the ground is challenging due to large variations of wind speed and direction. Proxies involving total reactive nitrogen $NO_y$ and ammonia ($NH_3$) relative to carbon monoxide (CO) can only be used to qualitatively evaluate smoke age due to large uncertainties of source emission factors and complexity caused by photochemistry (Selimovic et al., 2019; Kleinman et al., 2007). In contrast, the concentration ratio between $PM_{2.5}$ and CO ($PM_{2.5}/CO$) has shown potential for estimating smoke age (Yokelson et al., 2009; Selimovic et al., 2020). In this work, we determined the smoke conditions (young vs aged) primarily by comparing the field $\delta^{18}O$-HONO results with that obtained in our previous lab study that represents fresh emissions, with additional evaluation involving $\delta^{15}N$-HONO and relative concentration of HONO and $NO_2$ (Fig. 2). Note young and aged smoke refers





to negligible and large proportion of secondarily produced HONO respectively. We also take into account smoke sampling locations (i.e., approximate distance from the wildfire) to confirm the smoke age estimate. In brief, largely elevated $\delta^{18}$O-HONO obtained from field samples compared with that from the lab-controlled fire signifies significant atmospheric processing, and this will be discussed in detail below. Our grouping method using $\delta^{18}$O-HONO shows fairly consistent results with that derived from PM$_{2.5}$/CO for WFF and NF fire plumes (Kaspari et al., 2021). In addition to distinguishing

aged smoke from young smoke, the grouped $\delta^{18}$O and $\delta^{15}$N also allow us to characterize potential mechanism of secondary HONO formation in the aged smoke as well as NO-to-NO$_2$ oxidation pathways, with the HONO budget evaluated using the synergistic measurement of HONO, NO and NO$_2$ concentrations in the field.

## 2 Methodology

### 2.1 Description of Mobile Laboratory platforms: Molab and MACH-2

During the WE-CAN campaign in August of 2018, we conducted our measurements and sampling using the NOAA Chemical Science Division mobile laboratory (Molab), which was a cargo van with all instruments mounted on it. Meteorological instrumentation on the roof of the Molab provides temperature, relative humidity, wind speed, wind direction, altitude, and GPS coordinates. All additional instruments were mounted to the interior floor and ambient air is sampled through 1- or 2-meter Teflon inlets that exit the roof of the Molab via bored holes. NO and NO$_x$ concentrations were

measured with a Thermo Scientific Model 42i chemiluminescence NO/NO$_x$ analyzer owned by Brown University, with ±0.4 ppbv precision and 0.2 ppbv zero noise at 1 minute time resolution. Note the NO$_x$ concentration measured using the chemiluminescence analyzer serves as estimates due to known interferences from NO$_y$ species, mainly HONO and PAN. However, these data provide an upper limit of NO$_x$ level that supports the isotopic collections of NO$_x$, HONO and nitrate. HONO and HNO$_3$ concentrations were measured using University of New Hampshire's dual mist chamber/ion

chromatograph (MC/IC) system with an uncertainty of 3% at 5-minute resolution (Chai et al., 2019; Scheuer et al., 2003). During the FIREX-AQ field campaign in July-August of 2019, we mounted our sampling instruments in the NASA Langley mobile aerosol characterization platform (MACH-2) (Kaspari et al., 2021).

### 2.2 Description sampling location and strategy


While our sampling strategy was similar in both years, the actual sampling approach differed in response to fire condition and accessibility to fresh smoke from the mobile platforms.

During the 2018 WE-CAN campaign, our ground measurement and sampling targeted smoke from Rabbit foot fire (RF) in

the Challis area of Salmon-Challis National Forest in central Idaho, from August 9$^{th}$ to August 18$^{th}$ 2018 (Salmon-Challis





National Forest, 2018). Measurements during WE-CAN were made at various locations around the Challis area of Idaho impacted by the RF fire, consisting of 4 different conditions: young smoke during nighttime (YN), young smoke during daytime (YD), aged smoke during nighttime (AN), aged smoke during daytime (AD), and mixed daytime smoke (M) that contains smoke contributed by either night smoke or fresh smoke. To sample the young smoke, we drove the Molab to

Morgan Creek Road (MCR), which extends into a valley that was several kilometres away from the edge of the fire. We observed heavy smoke that based on distance and wind speed was expected to transport from the RF fire burning locations to the valley within a few hours or less. Three night-trips and two day-trips were made to MCR. While the nighttime measurements were conducted while driving the daytime work was carried out while parked at a spike camp (i.e. a campsite for firefighters and support personnel) at the upper end of MCR; the spike camp was about two kilometres from the fire,

which we were able to see while conducting the measurements. The aged smoke was sampled at three stationary sites located around the Challis area, each less than 30 km away from the RF fire. All of these sites were recreational vehicle parks that allowed for power plugins. A total of 7 nights and 4 days were measured. The sampling locations and driving map are shown in Fig. S1 with detailed information on the measurements listed in Table 1.

During 2019 FIREX-AQ, we investigated five wildfires in the western US including Shady fire (Idaho), Black Diamond fire (Montana), Williams Flats fire (Washington), Nethker fire (Idaho), and Little Bear fire (Utah) from Jul 24 to Aug 22 of 2019, and we intensively sampled the emissions from Williams Flats fire and Nethker fire based on the large size and easy access to sampling locations (Fig. S2). Similar to the 2018 field campaign, the measurements were conducted under YN, YD, AN, AD and M conditions.


**2.3 Collection of HONO, NO$_x$ and nitrate for isotopic analysis**

Nitrogen oxides (NO$_x$ = NO + NO$_2$), nitrous acid (HONO), particulate nitrate (p-NO$_3^-$) and nitric acid (HNO$_3$) were captured in the field using recently developed methods and sent to Brown University for analyses of isotopic composition (Chai et al.,

2019; Fibiger and Hastings, 2016; Chai and Hastings, 2018; Fibiger et al., 2014). In brief, HONO was completely captured at a pumping flow rate of ~10L/min with an annular denuder system (ADS), comprised (in order) of a Teflon particulate filter to remove p-NO$_3^-$, a Nylasorb filter to remove HNO$_3$, followed by two annular denuders, each coated with a premixed Na$_2$CO$_3$−glycerol−methanol−H$_2$O solution following a standard EPA method (Chai and Hastings, 2018). Within 24 hours after each collection, the coating was extracted in 10 mL of ultrapure water (18.2 MΩ) in two sequential 5 mL extractions.

NO$_3^-$ on the upstream Millipore filters and HNO$_3$ from the Nylasorb filters, if there was any, were extracted by sonicating the filters in ~30 mL ultrapure H$_2$O (18.2 MΩ). Samples with [NO$_3^-$] > 1 μM were analysed for isotopic composition (concentration techniques detailed below).





The denuder extracted solution with a pH of ~10 was transported to Brown University for concentration and isotopic
analysis, which was completed within 1.5 months after the sampling. The timescales for sample extraction and isotopic
analysis preserve both the solution concentration and isotopic composition of HONO in the form of nitrite (Chai and
Hastings, 2018). The two-denuder set up allows for minimization of the interference for both concentration and isotopic
analysis from other N-containing species that could be trapped and form nitrite in residual amounts on the denuders,
especially $NO_2$. Note HONO levels were above the minimum detection limit (0.07 mM) and the breakthrough amount of
HONO threshold is far from being reached given the concentrations (Table 1), flow rate (~ 8 L/min) and collection times (2 -
12 hours). Isotopic analysis of nitrite required collection of a minimum amount of 10 nmol. $NO_x$ was completely collected in
a impinging solution containing 0.25 M $KMnO_4$ and 0.5 M NaOH which oxidizes NO and $NO_2$ to $NO_3^-$ by pumping sampled
air through a gas washing bottle at a flow rate of ~4L/min. Collection time for HONO ranged from 2-12 hours and that for
$NO_x$ ranged from 0.75 – 2.5 hours depending on their mixing ratios to make sure sufficient samples were captured against
blanks for isotopic analysis (Fibiger et al., 2014; Fibiger and Hastings, 2016; Wojtal et al., 2016). Particulate filters and
Nylasorb filters were collected over 7-12 hours due to low concentration of particulate nitrate and $HNO_3$.

The samples from each collection system were retrieved and processed following the procedures described in (Chai et al.
(2019) and Methods) in a timely manner. All treated samples from $NO_x$, HONO, p-$NO_3^-$ and $HNO_3$ collection and their
corresponding blanks were analyzed offline for concentrations of $NO_2^-$ and $NO_3^-$ with a WestCo SmartChem 200 Discrete
Analyzer colorimetric system. The reproducibility of the concentration measurements was ±0.3 μmol $L^{-1}$ (1σ) for $NO_2^-$ and
±0.4 μmol $L^{-1}$ for $NO_3^-$ when a sample was repeatedly measured (n = 30). A detection limit of 0.07 μmol $L^{-1}$ for $NO_2^-$ and
0.1 μmol $L^{-1}$ for $NO_3^-$ was determined, and no detectable nitrite or nitrate was found in the blank denuder coating solution,
whereas blank $NO_3^-$ concentrations of ~5 μM are typical for the $NO_x$ collection method (Fibiger et al., 2014; Wojtal et al.,
2016). We only report the samples whose concentrations were at least 30% above $NO_3^-$ present in the blank $KMnO_4$ solution
upon purchase to avoid increasing the error associated with the isotopic composition (Fibiger et al., 2014). Note that $NO_3^-$
concentration was measured on the ADS solutions to verify whether and to what extent $NO_2^-$ was oxidized to $NO_3^-$ on
denuder walls because the denitrifier method will convert both $NO_3^-$ and $NO_2^-$ to $N_2O$ for isotopic analysis (see below).

**2.4 Isotopic analysis**

The denitrifier method was used to complete nitrogen ($^{15}N/^{14}N$) and oxygen ($^{18}O/^{16}O$) isotope analyses of separate $NO_3^-$
samples converted from $NO_x$, and $NO_2^-$ samples converted from HONO by quantitative conversion to $N_2O$ by denitrifying
bacteria *P. aureofaciens* (Casciotti et al., 2002; Sigman et al., 2001). The isotopic composition of $N_2O$ is then determined by
a Thermo Finnegan Delta V Plus isotope ratio mass spectrometer at *m/z* 44, 45 and 46 for $^{14}N^{14}N^{16}O$, $^{14}N^{15}N^{16}O$ and
$^{14}N^{14}N^{18}O$, respectively. Sample analyses were corrected against replicate measurements of the $NO_3^-$ isotopic reference
materials USGS34, USGS35, and IAEA-NO-3 (Böhlke et al., 2003), and that of the $NO_2^-$ isotopic reference materials N7373



and N10219. Precisions for $\delta^{15}N-NO_x$, $\delta^{15}N-HONO$ and $\delta^{18}O-HONO$ isotopic analysis across each of the entire methods are $\pm 1.3‰$, $\pm 0.6‰$ and $\pm 0.5‰$ respectively (Chai and Hastings, 2018; Fibiger et al., 2014).

## 3 Results and Discussion

### 3.1 Concentrations of HONO and NO$_x$

Among the three fires, increased HONO concentrations were observed in young smoke during both night (0.2 – 2.0 ppbv) and day (2.5 ppbv), while HONO level is significantly lower in aged smoke during both night (0.06 – 1.0 ppbv) and day (0.05 – 0.6 ppbv) as shown in Fig. 2 (a). Although median values show young night and day are significantly higher than aged smoke day and night, there is significant overlap between young nighttime and aged day and night for WF and Nethker fires. These ppbv to sub-ppbv HONO concentrations can be a major OH source in the ground areas (e.g., national forest) that are impacted by wildfire. We also determined the molar ratio HONO/NO$_2$ from the concentrations for each sample (Fig. 3), and the values represent the upper bound of [HONO]/[NO$_x$] (e.g., Table 1(a)). Median ratios of [HONO]/[NO$_2$] for the five smoke conditions are 0.35 (YN), 0.12 (YD), 0.07 (AN), 0.09 (AD) and 0.04 (MD). The median ratios of [HONO]/[NO$_2$] for the young smoke (0.35 for YN and 0.12 for YD) fall in the range of fresh emissions measured in the lab (0.13-0.53), and the field (0.05-0.33) (Yokelson et al., 2009; Selimovic et al., 2020 and references therein). Our results for YN are also in agreement with airborne measurements (0.34±0.08) from the BB-Flux campaign that occurred in parallel with WE-CAN, but are lower than the WE-CAN airborne observation of 0.72±0.34 during the day (Theys et al., 2020; Peng et al., 2020). It is worth noting the majority of the WE-CAN airborne data overlap with the rest measurements; and the discrepancy originating from one fire may be attributed to the different transport dynamics of fresh plumes (Peng et al., 2020). The wildfire plume diffused to the nearby ground area is expected to be more diluted than that directly injected upwards during the day; thus, the loss of HONO due to photolysis in the ground plume and/or ground uptake is faster than that in the upper altitude dense plumes. In the aged smoke, [HONO]/[NO$_2$] are greatly reduced to median values of 0.05 and 0.07 observed for AN and AD, respectively, lower than the lab-derived range (Fig. 3). This indicates HONO was lost faster than NO$_x$; however, given the 10-20 minute lifetime of HONO against photolysis during the day and up to a couple of hours during the night (Nie et al., 2015), and considering aged smoke was sampled 10s of km from the fire, HONO levels may be maintained via secondary chemistry due to the high particle loadings and other terrestrial surface reactions in wildfire plumes *(Alvarado and Prinn, 2009)*. While the concentration data are valuable for the ground-based setting near the fires, considerable uncertainty exists in the rate coefficients of the heterogeneous processes in daytime, as well as the HONO and NO$_2$ uptake coefficient and surface area densities (Appendix A). This makes it challenging to quantify the relative contribution of each potential pathway to the observed HONO budget.

### 3.2 Isotopic signatures of HONO and NO$_x$





In the 2016 FIREX Fire Laboratory experiment, we obtained $\delta^{15}N$ of $NO_x$ and HONO, as well as $\delta^{18}O$ of HONO in direct emissions from controlled burning of various vegetation biomass representative of the western US (Chai et al., 2019). The

lab-based $\delta^{15}N$ and $\delta^{18}O$ results serve as source signatures of biomass burning (BB) emissions: $\delta^{15}N$-$NO_x$ (-4.3‰ to +7.0‰) and $\delta^{15}N$-HONO (-5.3‰ to +5.8‰) are derived from biomass N and the transformation in the combustion process, and $\delta^{18}O$-HONO (5.2‰ to 15.2‰) incorporates $\delta^{18}O$ of molecular oxygen and water via combustion reactions (Chai et al., 2019). In the field, we expect that once $NO_x$ and HONO are released and transported, atmospheric processing including photochemistry and nighttime chemistry would cause the isotopic composition of emitted $NO_x$ and HONO to change.

By directly comparing the field-measured $\delta^{18}O$-HONO with that measured from lab-controlled burning, we separate the data observed in young smoke from that in aged smoke. Very young smoke largely reflects fresh wildfire emissions without significant atmospheric processing, while aged smoke $\delta^{18}O$-HONO should deviate from the lab values due to the influence of secondary chemistry involving RN cycling. The $\delta^{18}O$-HONO of young nighttime smoke ranged from 4.8‰ to 32.3‰ with a median value of 19.0‰, while the value in a single young daytime sample was 25.6‰ (Fig. 2(b)). There is a major overlap

between the lab results and young nighttime smoke, but with some higher $\delta^{18}O$-HONO values in the field observations. These results suggest the HONO sampled in young smoke was dominated by primary BB emissions from the nearby wildfire, but included contributions of secondarily produced HONO. By contrast, $\delta^{18}O$-HONO is greatly elevated in aged smoke from all three fires both day and night. In addition, two aged smoke samples are labelled as mixed smoke because the collection interval included both sunlit and dark periods. The enrichment of $\delta^{18}O$-HONO (up to 78‰), regardless of location and time,

suggests that HONO in these conditions is produced by secondary chemistry involving NO, $NO_2$ and nitrate, which transfer high $\delta^{18}O$ values due to $O_3$ influence via photochemistry (Appendix B) (Thiemens, 2006; Michalski et al., 2003). The varying $\delta^{18}O$-HONO values reflect different oxidizing environments, i.e., NO-to-$NO_2$ conversion via $RO_2$ versus $O_3$. These branching ratios can be determined if we resolve the dominant pathways for HONO production.

$\delta^{15}N$-HONO in the young smoke ranges from -0.3‰ to +7.4‰ with a median value of 2.8‰ for YN, and +3.4‰ for YD,

whereas that in the aged smoke shows decreased median values -2.9‰ and -1.8‰ for AN and AD respectively. In addition, the daytime aged smoke exhibits the largest variability (Fig. 2(c)), and this likely reflects daytime HONO secondary chemistry. It is noted that $\delta^{15}N$-$NO_x$ and $\delta^{15}N$-HONO measured across the entire period of all three fires at ground level ranges from -4.3‰ to +8.7‰ and -6.7‰ to +7.4‰, respectively, with the majority overlapping with the corresponding ranges found in the Fire Laboratory experiment and no significant difference in mean values (p-value >0.5) (Chai et al., 2019;

Fibiger and Hastings, 2016). This consistency suggests $\delta^{15}N$ is a reliable tracker generally for BB derived $NO_x$ and HONO, although there is clear variability between the different smoke conditions that can refine our understanding of reactive N cycling.





Our prior lab-controlled burning study revealed a linear relationship between $\delta^{15}N$-HONO and $\delta^{15}N$-$NO_x$, with $\delta^{15}N$-HONO
slightly more negative than $\delta^{15}N$-$NO_x$ in fresh BB emissions (Chai et al., 2019). This $\delta^{15}N$ relationship is plotted as a solid
line together with all field observations to illustrate the potential influence of atmospheric processing on the $\delta^{15}N$ -HONO
and -$NO_x$ (Fig. 4). The plot can be sub-divided into three regimes. In regime I, we find all of the $\delta^{15}N$ of $NO_x$ and HONO in
young smoke from both daytime and nighttime. In this young smoke regime, more positive $\delta^{15}N$ than that of the rest of our
samples is found for both species and all samples concur with the $\delta^{15}N$ relationship found for fresh emissions (Fig. 4). This,
along with the low $\delta^{18}O$-HONO associated with these samples (Fig. 2(b)), confirms HONO is not significantly affected by
secondary chemical processing in the air mass captured from fresh smoke. Regime II is filled with the results of daytime
aged smoke ~30 km away from the RF fire; these results exhibited much more positive $\delta^{15}N$-HONO than $\delta^{15}N$-$NO_x$ by 3‰-
6‰, and the largest (positive) discrepancy from the BB $\delta^{15}N$ relationship line, as shown in the upper left region of Fig. 4.
The daytime aged smoke also exhibited the highest values of $\delta^{18}O$-HONO observed (Fig 2.). All samples of aged nighttime
smoke that were collected fall in regime III. While the majority of the regime III data fall within the 95% confidence interval
for the lab-based $\delta^{15}N$ relationship, there is a tendency for these samples to have $\delta^{15}N$-HONO that was more negative than
$\delta^{15}N$-$NO_x$ to different degrees of up to -8.7‰. In particular, we hypothesize that the combination of more negative $\delta^{15}N$-
HONO values and elevated $\delta^{18}O$-HONO indicate secondary production of HONO. We next explore quantitative use of
$\delta^{15}N$-$NO_x$, $\delta^{15}N$-HONO and $\delta^{18}O$-HONO to understand the isotopic shifts in terms of secondary chemistry involving reactive
nitrogen cycling.

**3.3 Isotopic mass balance modelling**

In aged smoke, the observed $\delta^{18}O$-HONO enhancement and shift of $\delta^{15}N$ values away from the $\delta^{15}N$ $NO_x$-HONO line, as a
result of fast reactive nitrogen cycling, would be expected to derive from the integrated kinetic isotopic fractionation
(expressed as enrichment factor $^{18}\varepsilon$ and $^{15}\varepsilon$) associated with each of the loss/production processes (Fig. 1) weighted by their
relative contribution to the budget. For $\delta^{18}O$-HONO, we also took into account transferring effect of oxygen from different
O-containing reactants that produce HONO (as explained in Appendix B). In order to elucidate the relative role each process
plays in the HONO budget, we constructed an isotopic mass balance model for $\delta^{15}N$ and for $\delta^{18}O$.

In aged smoke, a deviation in $\delta^{15}N$, represented as $\Delta\delta^{15}N_{HONO\text{-}NOx}$ (= $\delta^{15}N$-HONO $-$ $\delta^{15}N$-$NO_x$), is simulated following Eq.
(1), where f is the fraction of reaction i (R numbering in Fig. 1) to total loss (L) or production (P) of HONO. $\delta^{18}O$-HONO is
simulated following Eq. (2), where the change of $\delta^{18}O$-$HONO_{i,P}$ arises from, in addition to kinetic isotopic fractionation, the
transferring of $\delta^{18}O_{i,t}$ (Eq. (3)) in the reactant (OH, NO, $NO_2$, $H_2O$, and $NO_3^-$) to the product HONO, as HONO contains two
O atoms that may stem from more than one reactant (Appendix B). $\delta^{18}O$ of all possible reactions that produce HONO are
evaluated as tabulated in Table S1 in the supplement, to help determine $\delta^{18}O$ of NO, $NO_2$ and HONO. The isotopic


enrichment factors $^{15}\varepsilon$ and $^{18}\varepsilon$ associated with each of the reactions R1-R7 illustrated in Fig. 1 are computed via theoretical

principles, as none of these key parameters are currently available in literature (Appendix B).

$$\Delta\delta^{15}N_{HONO-NOx}=\sum_{i,L}(f_{i,L}\times\Delta\delta^{15}N_{i,L})+\sum_{i,P}(f_{i,P}\times\Delta\delta^{15}N_{i,P}) \qquad (1)$$

$$\delta^{18}O-HONO=\sum_{i,L}(f_{i,L}\times{}^{18}\varepsilon_{i,L})+\sum_{i,P}(f_{i,P}\times\Delta\delta^{18}O-HONO_{i,P}) \qquad (2)$$

$$\Delta\delta^{18}O-HONO_{i,P}=\delta^{18}O_{i,t}+{}^{18}\varepsilon_{i,P} \qquad (3)$$

### 3.3.1 Modelling of $\delta^{15}N$ of HONO and $NO_x$ in aged daytime and nighttime smoke

We first simulated $\Delta\delta^{15}N_{HONO-NOx}$ for both daytime and nighttime aged condition using this model. According to the

potential HONO-$NO_x$ chemistry in ground areas impacted by wildfire smoke plumes (Fig. 1), HONO is expected to be

predominantly lost to photolysis (R1) during the day. It is well known that HONO can be produced via gas-phase radical

recombination reaction between NO and OH (R2) (Platt et al., 1980). However, the rate of R2, calculated from the rate

coefficient, the typical daytime OH concentration (1-2×10$^6$ molecule cm$^{-3}$) (de Gouw et al., 2006) in biomass burning

plumes and our measured mean $NO_x$ concentration, can only account for up to 15% of the HONO production rate (Appendix

A and Table A1). Under a typical pseudo steady state approximation (d[HONO]/dt ≈ 0), additional sources of HONO must

be included to balance the HONO budget. Thus, we modelled three scenarios varying the relative contribution of R2 as 5%,

10% and 15%. With rapid photolytic loss, HONO has a lifetime nearly two orders of magnitude shorter than the lifetime of

NO in R2 as well as that of $NO_2$ in R3 and nitrate in R4 (Fig. 1); thus, the $\Delta\delta^{15}N_{HONO-NOx}$ is mostly sensitive to the change in

$\delta^{15}N_{HONO}$ immediately upon photolysis, but overall remains constant associated with R2-R4 within the timescale of HONO

photolysis. Thus, the daytime $\Delta\delta^{15}N_{HONO-NOx}$ for aged smoke was simulated as a function of remaining HONO fraction, $f_{rp}$,

as a result of photolysis, following a Rayleigh-type isotopic fractionation scheme (Fig. 5). Generally, $\Delta\delta^{15}N_{HONO-NOx}$ follows

an exponential increase as $f_{rp}$ decreases. In other words, as more photolysis occurs the difference in the remaining $\delta^{15}N$-

HONO and the $\delta^{15}N$-$NO_x$ increases, and this is driven by the negative value of $^{15}\varepsilon_1$ which tends to enrich $^{15}N$ in the HONO

reactant (R1). The simulation was carried out for two different sets of HONO production mechanisms, with HONO

photolysis being the dominant loss fate. With mechanism M1 (solid lines in Fig. 5), photo-induced surface $NO_2$-to-HONO

conversion (R3) is the major pathway in addition to gas-phase OH+NO (R2) to produce HONO. As $^{15}\varepsilon_2$ has a positive value,

larger R2 contribution leads to higher $\Delta\delta^{15}N_{HONO-NOx}$. With mechanism M2 (dashed line in Fig. 5), nitrate photolysis (R4) is

included in addition to R2 and R3 in the HONO production mechanism. Taking the contribution of R2 of 10% as a constant,

three scenarios were modelled by varying the relative contribution of R3 (75%-85%) and R4 (5%-15%). The results suggest

larger R4 contribution yields lower $\Delta\delta^{15}N_{HONO-NOx}$ due to severe $^{15}N$ depletion associated with nitrate photolysis ($^{15}\varepsilon_4 \leq -$

47.9‰) (Appendix B). Importantly, the addition of R4 in M2 also lowers $\Delta\delta^{15}N_{HONO-NOx}$ compared to M1. By applying the

field-observed $\Delta\delta^{15}N_{HONO-NOx}$ for the aged daytime smoke to the model, we solved $f_{rp}$ for all scenarios and plotted these as





circles in Fig. 5. All five daytime aged data from RF can be reproduced by M1 under all three scenarios; by contrast, via M2, none of the three scenarios can explain the two highest $\Delta\delta^{15}N_{HONO-NOx}$ observed in the field. As such, we conclude R4 plays a minor role (<5%) in the secondary HONO production in the aged daytime smoke. Rather, HONO forms primarily via R2 and R3 during the day in the areas impacted by aged wildfire smoke.

For the nighttime smoke, we calculated that the HONO budget is maintained by R5-R7 (Fig. 1). $\Delta\delta^{15}N_{HONO-NOx}$ reflects the combination of kinetic isotopic fractionation $^{15}\varepsilon_5$ associated with the HONO loss R5 and production reactions (R6 and R7 in proportion). With our calculated uptake $^{15}\varepsilon_5$ (-2‰), and estimated $^{15}\varepsilon_6$ or $^{15}\varepsilon_7$ (ranging from -2.9‰ to -4.5‰), we obtained $\Delta\delta^{15}N_{HONO-NOx}$ ranging from -0.9 to -2.5‰ when uptake and production occurs at a similar time scale (rate coefficient), and this can explain the majority of observed aged nighttime results (regime III, Fig. 4). Two aged nighttime points sampled for

RF (Aug 16 and 17, 2018) fall outside of the predicted range, with much lower $\Delta\delta^{15}N_{HONO-NOx}$ (-8.7‰ and -5.5‰ respectively). These two samples were associated with 2-10 times elevated $NO_x$ concentration compared to the previous 4 nights and likely higher concentrations of particulate matter (Fig 2(a); Fig. S3 in the supplement). This could cause an accelerated conversion of $NO_2$-to-HONO, which is not accounted for in the steady state estimation above, leading to the much lower $\Delta\delta^{15}N_{HONO-NOx}$ values that were observed.

**3.3.2 Modelling of $\delta^{18}O$ of HONO in aged daytime and nighttime smoke**

$\delta^{18}O$-HONO of daytime aged smoke was modelled following M1 (R1-R3) derived based upon the $\delta^{15}N$ modelling results: NO and $NO_2$ are cycled via $NO_2$ photolysis and NO oxidation by $O_3$ and/or peroxy radicals ($RO_2$ including $HO_2$) during the day, through which $\delta^{18}O$ of $O_3$ and $RO_2$ can be passed to NO and $NO_2$ via mass transfer (Eqs. (B9)-(B11)). $O_3$ is known to have an intrinsically high $\delta^{18}O$ value of up to ~117‰ caused by unique isotopic fractionation associated with photochemical

gas-phase $O_3$ formation (Thiemens, 2006), while OH and $RO_2$ have very low $\delta^{18}O$ values (Thiemens, 2006). $O_3$ participation in reactive N cycling involving $NO_x$ (R8) results in high $\delta^{18}O$ of $NO_2$ (Michalski et al., 2003; Walters et al., 2018). In pseudo photochemical steady state, NO and $NO_2$ are expected to have similar $\delta^{18}O$ that is a result of competition between $O_3$ and $RO_2$ oxidation (R8-R10), expressed as $f_{O_3/RO_2}^{NO}$ via Eqs. (4) and (5) below.

$$NO + O_3 \rightarrow NO_2 + O_2 \hspace{8cm} R8$$

$$NO + HO_2/RO_2 \rightarrow NO_2 + HO/RO \hspace{6cm} R9a$$

$$\rightarrow HONO_2/RONO_2 \ (5\%) \hspace{6cm} R9b$$

$$NO_2 + h\nu + O_2 \rightarrow NO + O_3 \hspace{7cm} R10$$

$$\delta^{18}O\text{-}NO \approx \delta^{18}O\text{-}NO_2 = f_{O3/(O3+RO2)}^{NO} \times \delta^{18}O\text{-}O_3 + (1 - f_{O3/(O3+RO2)}^{NO}) \times \delta^{18}O\text{-}RO_2 \hspace{2cm} (4)$$



$$f^{NO}_{O3/(O3+RO2)} = \frac{k_{NO+O3}[O_3]}{k_{NO+O3}[O_3] + k_{NO+RO2}[RO_2]}$$ (5)

The $\delta^{18}O$ signature is subsequently passed to HONO when it is produced from NO (R2) and $NO_2$ (R3) during the day and from $NO_2$ (R6 and R7) during the night, and thus $\delta^{18}O$-HONO is a positive linear function of $f^{NO}_{O3/(O3+RO2)}$ if kinetic isotopic fractionation ($^{18}\varepsilon$) associated with these processes are fixed values (as calculated in Appendix B). Given that HONO is

predominantly produced via R2 and R3 in aged daytime smoke (Fig. 5), $\delta^{18}O$-HONO was simulated following the three M1 scenarios with the contribution of R2 varying from 5% to 15%. All three scenarios reproduced the range of our field results for aged daytime smoke, further pointing to M1 as explaining the HONO in this environment. In addition, the variation of $\delta^{18}O$ was driven by differing oxidation that is determined by $f^{NO}_{O_3/RO_2}$, which depends on the relative concentration of $O_3$ to $RO_2$ (Figs. S4 and S5). $f^{NO}_{O_3/RO_2}$ corresponding to each observed $\delta^{18}O$-HONO were solved and plotted in Fig. 6(a). We found

$f^{NO}_{O_3/RO_2}$ decreased by less than 0.02 as the contribution of R2 to total HONO production decreased from 15 to 5%. On the other hand, $\delta^{18}O$-HONO changes sensitively with varying $f^{NO}_{O_3/RO_2}$, increasing from 50.2‰ to 78.0‰ as the fraction of NO oxidized by $O_3$ rather than $RO_2$ increases from 0.34 to 0.65.

$\delta^{18}O$-HONO of nighttime aged smoke was modelled following the nighttime chemistry (R5-R7), i.e. taking $NO_2$ conversion

as the source and surface uptake as the sink. In areas impacted by nighttime aged smoke, HONO forms from wildfire derived $NO_2$ residing in the nocturnal boundary layer. As the two pathways (R6 and R7) for heterogeneous $NO_2$ conversion lead to very different $\delta^{18}O$-HONO stemming from different $\delta^{18}O$ transfer (Appendix B), we examined the relative importance of the two pathways for HONO production by varying the relative contribution between the two pathways and comparing to the observed $\delta^{18}O$-HONO (Fig. 6(b)). If HONO is constrained to exclusively form via R6 (surface hydrolysis), the model would

require an unrealistic $f^{NO}_{O_3/RO_2}$ >100% to explain $\delta^{18}O$-HONO > 55‰. Even for samples with lower $\delta^{18}O$-HONO values (34‰-52‰), the high branching ratio $f^{NO}_{O_3/RO_2}$ (> 0.6) required to create such large enrichment is unrealistic for BB environments. In particular, $[O_3]/[RO_2]$ converted from $f^{NO}_{O_3/RO_2}$ solved under this mechanism (0.24 to 0.72) is at least twice as large as values derived from the previous field measurement of aged wildfire smoke (Baylon et al., 2018). By contrast, inclusion of R7 in addition to R6 in rate ratios 3:1 and 20:1 based on previous lab studies (Kebede et al., 2016; Scharko et al., 2017) can elevate

the modelled $\delta^{18}O$-HONO and explain all observed $\delta^{18}O$-HONO values. This suggests $NO_2$-to-HONO heterogeneous conversion catalysed by surface-hosted iron oxides and quinone (R7) in the nighttime aged smoke proceeds significantly faster than $NO_2$ hydrolysis (R6). Our isotopic analyses provide evidence for participation of such pathway in BB environments, and also shows the capability to constrain the relative importance between these two pathways. Although the daytime $\delta^{18}O$-HONO can be larger than that of nighttime aged smoke, similar $[O_3]/[RO_2]$ ratios are derived from our solved





$f_{O_3/RO_2}^{NO}$ and are consistent with the limited field measurements (Parrington et al., 2013; Baylon et al., 2018), and further indicate the important role peroxy radicals play as an oxidant in wildfire smoke impacted environments.

## 4 Conclusion

As wildfire has enormously impacted climate, air quality and ecosystems in the past and is expected to worsen (Westerling, 2016), accurately tracking wildfire derived reactive nitrogen species (i.e. $NO_x$ and HONO) and their cycling is extremely important for quantifying and mitigating key pollutants such as $O_3$ in wildfire impacted areas both close to the fire and thousands of kilometres downwind. We show $\delta^{15}$N-HONO and $\delta^{15}$N-$NO_x$ can serve as a powerful tool to track BB sources and constrain secondary HONO production pathways. With the help of field-observed $\delta^{18}$O-HONO, we grouped our

measured relationship between the $\delta^{15}$N-HONO and $\delta^{15}$N-$NO_x$ into three different regimes, which clearly distinguish among young wildfire plume, aged daytime plume as well as aged nighttime plume. The $\delta^{15}$N results allow for constraining the daytime HONO budget particularly secondary production mechanisms via the isotope mass balance simulation. The use of excess $\delta^{15}$N ($\Delta\delta^{15}$N$_{HONO\text{-}NOx}$) also provides an approach for constraining HONO budgets in other environmental settings, such as urban ambient areas and remote areas including forest and polar regions. Furthermore, by combining $\delta^{15}$N emission

source signatures and chemical fractionation characteristics, we could potentially track the impact and relative role of wildfire derived reactive nitrogen more extensively when the plume transfers thousands of kilometres downwind and mixes with other air such as urban plumes. In addition, the $\delta^{18}$O-HONO results not only offer direct evidence for secondary production of HONO that allows for determination of the NO oxidizing branching ratio between $O_3$ and $RO_2$, but also constrains nighttime HONO production mechanisms. We expect to apply the $\delta^{18}$O-HONO approach to a variety of

atmospheric settings for constraining the HONO budget and its cycling with other reactive nitrogen species as well as $O_3$. As such, online isotopic measurement techniques with higher time resolution will benefit the use of stable isotopes and broaden its application in atmospheric chemistry. In the meantime, in order to more accurately quantify the relative contribution of these potential pathways, further experimental and theoretical investigations on isotopic fractionation characteristics of each pathway under various environmental conditions are required.


## Appendix A. Challenges of HONO budget estimation under different conditions based upon concentrations

A common approach to quantitatively understand the wildfire-derived HONO budget—its direct emissions, secondary productions and sinks—is to use concentration-based mass balance calculation. Ideally, if we know the rate coefficients and

reactant concentrations for each of the pathways, we would be able to quantify the relative contribution of each pathway to



the total HONO concentration measured in the field under the assumption of pseudo steady-state approximation (PSSA) as described in Eq. (A1). In aged smoke, we expect HONO is almost exclusively produced from secondary formation. During the day, HONO is predominantly lost to photolysis with a coefficient depending on solar zenith angle differing with time of the day, while one or more reactions of R2-R4 may be responsible for producing HONO (Fig. 1). Under PSSA, using the

well quantified rate coefficient $k_2$, observed NO and HONO concentrations, estimated OH concentration, and TUV model calculated HONO photolysis coefficient $j_{HONO}$, we estimated $P_{OH+NO}$ via Eq. (A2) and found R2 can only contribute 2%-15% (Table A1) of the total HONO production under the ambient conditions when the five aged-day samples were collected. This suggests at least 85% of HONO was produced from heterogeneous HONO formation via R3 and/or R4.

$$\frac{d[HONO]}{dt} = R_{emission} + R_{production} - R_{loss} \approx 0 \qquad (A1)$$

$$P_{OH+NO} = \frac{k_2 [OH][NO]}{j_{HONO} [HONO]} \qquad (A2)$$

HONO production from photo-enhanced $NO_2$ conversion has been proposed to take place on various types of surfaces. However, the uptake coefficient ($\gamma_{NO2 \rightarrow HONO}^{hv}$), which indicates the probability of $NO_2$ collisions with a surface that results in formation of a HONO molecule, varies by at least three orders of magnitude depending on the specific type of surface materials. For instance, $\gamma_{NO2 \rightarrow HONO}^{hv}$ on soot particles was found to range from $3.7 \times 10^{-4}$ to $1.1 \times 10^{-3}$ s$^{-1}$ (Ammann et al., 1998),

while that on surfaces comprised of humic acid was measured as $2-8 \times 10^{-5}$ s$^{-1}$ in several lab studies (Stemmler et al., 2006; Scharko et al., 2017). The latter is consistent with daytime modeling results of $6 \times 10^{-5}$ s$^{-1}$ (Wong et al., 2013). Additionally, much smaller ($10^{-7}$ -$10^{-6}$ s$^{-1}$) $\gamma_{NO2 \rightarrow HONO}^{hv}$ was obtained for metal oxide surface such as $TiO_2/SiO_2$ (Ndour et al., 2008).

Daytime photolysis of nitrate ($HNO_3$ and $pNO_3^-$) via R4 has also been proposed as an important renoxification pathway that

produces HONO and $NO_2$ in low $NO_x$/remote environments (Zhou et al., 2011) as well as high $NO_x$/urban settings with abundant urban grime (Baergen and Donaldson, 2016, 2013). The $p$-$NO_3^-$ and surface-adsorbed $HNO_3$ were found to be photolyzed with rate coefficients 2-3 orders of magnitude larger than gas-phase $HNO_3$, and possess lifetimes of as low as a few hours  (Ye et al., 2017). However, the rate coefficient of R4 is poorly constrained. Not only have the branching ratio between $NO_x$ producing channel and HONO forming channel been poorly known (Baergen and Donaldson, 2016), but

previous laboratory measured nitrate photolysis rate coefficients also vary by up to 3 orders of magnitude (Ye et al., 2017). The uncertainty is even greater when it is complicated with the dependence on relative humidity, particle composition and pH.

During the night, HONO is primarily lost to uptake on surfaces including aerosols and soils and the uptake coefficient can be

expressed by Eq. (A3)

$$L_{HONO}^{uptake} = 0.25 \times \gamma_{HONO} \times \overline{\omega_{HONO}} \times S/V \times 100 \qquad (A3)$$





In this equation, $\overline{\omega_{HONO}}$ is the mean thermal HONO molecular velocity calculated by

$\overline{\omega_{HONO}} = \sqrt{8RT/\pi M}$ , where R, T, and M are the gas constant, absolute temperature and molecular weight. S/V is the surface-to-volume ratio ($cm^2/cm^3$). The uptake coefficient $\gamma_{HONO}$ was measured to be $10^{-5}$ for soil surface and in the range of $10^{-5}$-$10^{-3}$ for aerosol particle surface (Donaldson et al., 2014b; Wong et al., 2012). In addition, OH+HONO occurs at rates 1-2 orders of magnitude smaller than the uptake and therefore plays a minor role. The combined loss processes lead to a HONO lifetime of about 4 hours during the night.


HONO is generally assumed to be produced via heterogeneous $NO_2$ hydrolysis disproportionation (R6) (Finlayson-Pitts et al., 2003), and the production rate of HONO is estimated by Eq. (A4), expressed in the unit of ppbv-HONO $ppbv^{-1}$-$NO_2$ $s^{-1}$.

$$P_{HONO}^{night} = 0.5 \times R_{NO2 \to HONO}^{surface} = 0.5 \times \gamma_{NO2} \times \overline{\omega_{NO2}} \times S/V \times 100 \quad (A4)$$


where $\overline{\omega_{NO2}}$ is the mean $NO_2$ molecular velocity, S/V is the surface-to-volume ratio of particles, which could range from 9.0 $\times$ $10^{-6}$ $cm^2/cm^3$ to 3.0 $\times$ $10^{-4}$ $cm^2/cm^3$ for normally polluted areas and highly polluted areas respectively (Spataro and Ianniello, 2014). The S/V in biomass plume has huge uncertainty; additionally, ground surface may also play a role in nighttime HONO production(Scharko et al., 2017; Kebede et al., 2016; Stemmler et al., 2006), however, its S/V is not well

defined/quantified.

Overall, considerable uncertainty remains regarding the rate coefficient of the heterogeneous processes in the daytime, as well as the HONO and $NO_2$ uptake coefficients and S/V ratio. This uncertainty, complicated further with large variability of fire behavior and emissions, make the HONO budget quantification extremely challenging.

**Appendix B. Quantification of isotopic fractionation factor**

**B.1 Nighttime processes**

**B.1.1 Isotopic fractionation of N and O associated with nighttime uptake**

Surface uptake is the major sink for HONO during the night. Surface uptake of HONO has been found to be kinetically limited by bulk diffusion in particles containing viscous organic-water matrices, and incorporate two simultaneous processes:

1) reactive uptake of HONO on the bare particle/minerals surface, and 2) accommodation and reaction of HONO in the bulk aqueous layer, that is affected by pH and diffusion in the organic-water matrix (Donaldson et al., 2014b). The uptake coefficient of HONO is determined by the competition between these two processes as a function of fraction of water coverage on the surfaces $\theta_{H2O}$ ranging from 0 to 1 in Eq. (B1), where $\gamma_0$ and $\gamma_l$ are the reactive uptake coefficients of HONO onto particle (mineral/soil) surfaces at dry ($\theta_{H2O}$=0) and wet ($\theta_{H2O}$=1) conditions, respectively. Under completely dry



conditions ($\theta_{H2O} = 0$ or relative humidity (RH) = 0%), the former process is dominant, and the isotopic fractionation can be estimated by the ratio of square root of inverse mass, which is caused by different thermal velocities ($\omega_{HONO}$) of two isotopologues following Eq. (B2), where R is the gas constant, T is absolute temperature and M is the molecular weight. Thus, heavier isotopes are depleted in HONO, resulting in -10‰ and -20‰ for $^{15}\varepsilon$ and $^{18}\varepsilon$ respectively. By contrast, under wet conditions when RH is 30% which results in a monolayer water coverage on particle surfaces ($\theta_{H2O}=1$), the aqueous

layer uptake becomes dominant and the wet uptake coefficient $\gamma_l$ can be mechanistically simulated with a resistor model simplified as Eq. (B3) (Hanson, 1997; Pöschl et al., 2007). In Eq. (B3), α is the accommodation coefficient describing the probability that a HONO molecule striking water-coated particle enters into the bulk liquid phase, and $\Gamma_b$ is the solubility of HONO in the bulk water in the particles or soils. $\Gamma_b$ can be calculated with Eq. (B4), where $D_a$ is the apparent diffusion coefficient of HONO in the particle-water (soil(organics)-water) matrix, and τ is the exposure time. $H_{eff}$ is the effective

Henry's law constant that depends on the absolute Henry's law constant for HONO, pH, and acid dissociation constants for HONO ($K_{a1}$) and H$_2$NO$_2^+$ ($K_{a2}$).

$$\gamma_{HONO} = \left(1 - \theta_{H_2O}\right)\gamma_0 + \theta_{H_2O}\,\gamma_l \tag{B1}$$

$$\omega_{HONO} = \sqrt{\frac{8RT}{\pi M}} \tag{B2}$$

$$\gamma_l = \frac{1}{\alpha} + \frac{1}{\Gamma_b} \tag{B3}$$

$$\Gamma_b = \frac{4H_{eff}RT}{\omega_{HONO}}\sqrt{\frac{D_a}{\pi\tau}} \tag{B4}$$

$$k_{u-HONO} \propto \frac{\gamma_l \times \omega_{HONO}}{4} \tag{B5}$$

Taking the previously measured HONO $\gamma_l$ of $2\times10^{-5}$ as that for the light isotopologue, and α of $5.8\times10^{-5}$ as a constant (Donaldson et al., 2014b), $\Gamma_b$(H$^{16}$O$^{14}$N$^{16}$O) is calculated to be $1.36\times10^{-5}$ following Eq. (B3). As derived from Eqs. (B2) and

(B4), $\Gamma_b$ ratio between two isotopologues equals the ratio between the two molecular weights, and therefore $\Gamma_b$(H$^{16}$O$^{15}$N$^{16}$O) and $\Gamma_b$(H$^{16}$O$^{15}$N$^{16}$O) were calculated and used to derive the corresponding $\gamma_l$ values. The fractionation factor associated with HONO uptake ($\alpha_{u-HONO}$), defined as the ratio between heavy and light rate coefficients ($k_H/k_L$), were calculated following the relationship determined by Eq. (B5). On the basis of this model, we estimate that the isotopic fractionation associated with the wet uptake process to be -2‰ and -4‰ for $^{15}\varepsilon$ and $^{18}\varepsilon$ respectively. From our calculation, RH clearly influences isotopic

fractionation in the range of 0-30%, with wet uptake of HONO favouring a smaller kinetic isotope effect than dry uptake.





**B.1.2 Isotopic fractionation of N and O associated with each nighttime HONO production pathway**

Heterogeneous conversion of $NO_2$ to HONO has been widely accepted as the major secondary HONO production source during the night. However, the mechanism via which the conversion occurs remains disputed. Additionally, the kinetic isotopic fractionation factor (KIF) associated with this process has never been measured or calculated. $NO_2$ hydrolysis (R6) on a variety of surfaces was determined to be a major source of HONO production. A compelling mechanism proposed by Finlayson-Pitts (Finlayson-Pitts et al., 2003) suggests R6 consists of a series of key steps including 1) dimer $N_2O_4$ formation from recombination of two $NO_2$ molecules in the gas phase, and uptake of gaseous $N_2O_4$ by thin water film on the top

surface layer, 2) aqueous phase isomerization of symmetric $N_2O_4$ to asymmetric $ONONO_2$ which is subsequently autoionizing to $NO^+NO_3^-$ and reacting with $H_2O$ to form HONO and $HNO_3$, and 3) desorption of HONO from aqueous to gas phase. Recently it was shown that reduction of $NO_2$ on iron-bearing minerals and quinone-rich humic acid in soils and particulate matter (R7) leads to faster HONO production than $NO_2$ hydrolysis. Although differing in reaction mechanism, the two possible pathways (R6 and R7) proceed in three steps including uptake of $NO_2$ into surface aqueous layer, reactions in

aqueous phase, and desorption of HONO from aqueous to gas phase. The first two steps are limited by aqueous diffusion, and it is reasonable to assume diffusion-limited processes in the aqueous phase create no KIF. As HONO desorption may involve hydrogen bond breaking of complex $HONO\bullet\bullet\bullet(H_2O)_n$, this process likely determines the KIF associated with the heterogeneous $NO_2$-to-HONO conversion ($\alpha_d$), as calculated by Eq. (B6), where $\mu_l$ and $\mu_h$ are the reduced mass for the light and heavy isotope containing pair, respectively (Shi et al., 2019). As a result, $^{15}\varepsilon$ and $^{18}\varepsilon$ are estimated to be -2.9‰ (n=1) to -

4.5‰ (n=2) and -5.7‰ (n=1) to -8.9‰ (n=2) respectively. For the isotope mass balance modelling, mean values of $^{15}\varepsilon$ (3.7‰) and $^{18}\varepsilon$ (7.4‰) were generally used for steady-state R3, R6 and R7 under steady-state condition, and the low (n=2) and high (n=1) values were used to evaluate the lower and upper bound.

$$\alpha_d = \sqrt{\mu_l/\mu_h} \tag{B6}$$


**B.2 Daytime**

**B.2.1 HONO photolysis**

The isotopic effect associated with photolysis (PIE) of HONO is calculated for the first time, following the $\Delta$ZPE-approach

proposed by Yung and Miller (1997) to determine the PIE of $N_2O$ photolysis. In principle, the absorption spectrum for the same kind of electronic transition is expected to be similar in shape and intensity upon isotopic substitution, based on the assumption that the electronic potential energy surface is constant for each isotopologue. This assures the continuum levels (leading to photolysis) of the excited state are not significantly changed while the vibrational levels of the ground state vary





with isotopic substitutions due to mass difference. The latter results in a lower ground state zero point energy (ZPE) for a heavy isotopologue than a light one, and cause blue shift in the absorption spectrum of the heavy isotopologue relative to the light one (Miller and Yung, 2000). When exposed to sunlight in the troposphere (>290 nm), HONO is known to feature a set of progressive absorption bands between 310-370 nm arising from electronic excitation $\widetilde{X}^1A' \rightarrow \widetilde{A}^1A''$, which result in HONO photolysis to OH and NO with nearly unity quantum yield (Cox et al., 1980; Suter and Huber, 1989). Under the aforementioned assumptions, we calculate the spectra blue shift of all three heavy isotopologues (HO$^{15}$NO, H$^{18}$ONO or

HON$^{18}$O) relative to that of HONO using the ΔZPE-approach as shown in Fig. B1 and Tables B1 and B2.

We calculated ΔZPE from $1/2\ \Sigma\Delta v_i$, where $\Delta v_i$ is the ground state vibrational frequency difference between the normal isotopologue (HONO) and the heavier isotopologue (HO$^{15}$NO, H$^{18}$ONO or HON$^{18}$O) for each vibrational mode calculated via forced field by Monse et al. (MONSE et al., 1969). Note only HO$^{15}$NO UV absorption was measured in previous study that reported an average blue shift of ~20 cm$^{-1}$ (8-40 cm$^{-1}$) relative to HONO, and this is consistent with our calculation

(Table B1). Note trans-HONO/cis-HONO abundance ratio is 2.5 at room temperature (Suter and Huber, 1989), and the difference of ΔZPE for t-HONO and c-HONO are less than 0.5% for $^{15}$N, and 2% for $^{18}$O (Table B1). The effect of the difference on j calculation is negligible. With the measured absorption cross-section of HONO between 293-400 nm and the quantified blue shift of all three isotopologues, we calculate each photolysis rate coefficient following Eq. (B7), which is the integral of photolysis quantum yield $\Phi_a(\lambda)$ ($\approx$1), absorption cross-section $\sigma_a(\lambda)$ and solar actinic flux $I(\lambda)$ as a function of

wavelength. $I(\lambda)$ is computed with the radiation transfer model TUV (http://www.acd.ucar.edu/TUV, Madronich and Flocke, 1998)) at various locations and time during our sampling period. With these j values listed in Table B2 ($j$, $j_{15N}$, $j_{18O1}$, $j_{18O2}$), the fractionation constant ($^{15}\varepsilon$ and $^{18}\varepsilon$, ‰) associated with HONO photolysis is calculated following Eq. (B8), where $j'$ and $j$ are photolysis rate coefficient of heavy and light isotopologues respectively. Note we take the average of $j_{18O1}$ and $j_{18O2}$ as $j_{18O}$ assuming the $^{18}$O is equally distributed between the two O-sites of HONO. Results shows $^{15}\varepsilon$ and $^{18}\varepsilon$ ranges from -1.9‰

to -4.3‰ (mean = 3.0‰, 1σ = 0.7‰, n =18) and -1.9‰ to -5.9‰ (mean = 3.1‰, 1σ = 1.0‰, n =18) respectively when HONO photolysis rate decreases from $1.4\times10^{-3}$ s$^{-1}$ to $5.3\times10^{-4}$ s$^{-1}$.

$$j = \int \sigma_a(\lambda)\ \Phi_a(\lambda)\ I(\lambda)\ d(\lambda) \qquad (B7)$$

$$\varepsilon = \left[\left(\frac{j'}{j}\right) - 1\right] \times\ 1000\ ‰ \qquad (B8)$$

The negative values of $^{15}\varepsilon$ and $^{18}\varepsilon$ suggest both $^{15}$N and $^{18}$O will be enriched in the remaining HONO upon photolysis.

Applying a Rayleigh fractionation model described by equation Eq. (B9), we obtain δ$^{15}$N and δ$^{18}$O of HONO ($\delta_f$) as a function of the fraction of HONO left after photolysis ($f$). The initial isotopic composition of HONO ($\delta_0$) is taken from nighttime young smoke mean values in Table 1, as they are the best estimate of the fresh emission from the fires we investigated.

$$\ln(\delta_f + 1000‰) =\ \varepsilon \ln(f) + \ln (\delta_0 + 1000‰) \qquad (B9)$$





### B.2.2 Isotopic fractionation of N and O associated with each daytime HONO production pathway

OH + NO (R2) is a radical-radical recombination reaction, which is characteristic of stabilization of activated complex HONO* via collisional energy transfer. This reaction type is characteristic of large KIF that enriches heavier isotopologues in the product at the low-pressure limit but almost no KIF at the high-pressure limit. The closer a reaction system is to the high-pressure limit, the less fractionation occurs (Chai and Dibble, 2014). Under the atmospheric pressure, the rate coefficient $k_1$ is in the fall-off region but close to the high-pressure limiting rate coefficient $k_{atm} = 1/3 \ k^{\infty}$ (Forster et al., 1995). Therefore, we expect a moderate positive $^{15}\varepsilon$ (~10‰) and $^{18}\varepsilon$ (~15‰) (Chai and Dibble, 2014; Burkholder et al., 2019).

Kinetic isotopic fractionation (KIF) associated with photo-enhanced $NO_2$ conversion is not known. Similar to the nighttime heterogeneous $NO_2$ conversion, R3 is also expected to occur in the surface aqueous phase and the overall KIF is largely determined by that associated with the desorption of HONO from aqueous to gas phase. Thus, $^{15}\varepsilon_3$ and $^{18}\varepsilon_3$ are the same as that of R6 and R7 (Appendix B).

KIF associated with $HNO_3$/p-$NO_3^-$ photolysis (R4) in the atmosphere has never been measured experimentally, and lack of
p-$NO_3^-$ absorption spectroscopy hinders calculation. $^{15}N$ enrichment factor ($^{15}\varepsilon$) for photolysis of snow surface-adsorbed $HNO_3$ under natural sunlight was theoretically determined to be $\leq$ -47.9‰ following the $\Delta$ZPE approach Yung and Miller (1997), which well explained the $^{15}\varepsilon$ laboratory-measured for snow surface nitrate photolysis under the radiation of simulated sunlight (Berhanu et al., 2014; Frey et al., 2009). If we take this $^{15}\varepsilon$ value, and the measured $\delta^{15}N$ of nitrate (8‰ to 20‰), the HONO produced from surface nitrate photolysis will be very negative (-38.9 to -27.5‰) within 2 hours of photolysis. $^{18}O$
enrichment factor ($^{18}\varepsilon$) for photolysis of snow surface-adsorbed $HNO_3$ has been measured to range from 6.0‰ to 12.5‰ (Frey et al., 2009; Berhanu et al., 2015).

### B.3 $\delta^{18}O$ transferring coefficient by different pathways

For $^{18}O$, in addition to KIF (enrichment factor, $\varepsilon_O^i$ in ‰), $\delta^{18}O$ transferring from different reactants greatly influence $\delta^{18}O$-HONO ($\delta^{18}O_{i,t}$), especially when the two O atoms of HONO are derived from different reactants. That is, HONO formed from different pathways (R2, R3, R6, R7) consists of $\delta^{18}O$ of each O-containing reacting partner in proportion determined by stoichiometry of reaction i, expressed with Eqs. (10)-(12). In R2, OH and NO equally contribute their O-atom to HONO expressed with Eq. (10); In R3 and R7, $NO_2$ is the exclusive O source of HONO while $H_3O^+$ only contribute a $H^+$ to HONO
(Ammann et al., 1998; George et al., 2005; Stemmler et al., 2006; Scharko et al., 2017; Kebede et al., 2016); In R6, the hydrolysis mechanism discussed in Appendix B suggests the $H_2O$-derived $OH^-$ and $NO_2$-derived $NO^+NO_3^-$ equally contribute their O-atom to HONO (Finlayson-Pitts et al., 2003).

$$\delta^{18}O_{2,t} = \frac{1}{2}\delta^{18}O\text{-}OH + \frac{1}{2}\delta^{18}O\text{-}NO \tag{B10}$$





$\delta^{18}O_{3(or\ 7),t}=\delta^{18}O\text{-}NO_2$                                                                                     (B11)

$\delta^{18}O_{6,t}=\frac{1}{2}\delta^{18}O\text{-}H_2O + \frac{1}{2}\delta^{18}O\text{-}NO_2$                                              (B12)

During the day, NO-NO$_2$ equilibrium is maintained via NO$_2$ photolysis and NO oxidation by O$_3$ and/or RO$_2$ following R8 and R9, and NO and NO$_2$ are expected to possess similar $\delta^{18}O$ and this can be expressed as $\delta^{18}O\text{-}NO_x$. During the night, due
to increased sink of NO$_x$ and decreased O$_3$ concentration, $\delta^{18}O\text{-}NO_x$ is expected to be lower than during daytime. NO$_x$ resulting from R8 and R9 should carry $\delta^{18}O$ of RO$_2$ and O$_3$ respectively via transfer; as RO$_2$ and O$_3$ have very different $\delta^{18}O$ values ~ +23‰ and +117‰ respectively, the competition between R8 and R9 critically affects $\delta^{18}O\text{-}NO_x$, as described by equations Eqs. (11) and (12). OH radical in the troposphere has been calculated to be -35‰ depleted in $^{18}O$ relative to H$_2$O as a result of isotopic exchange at 298 K (Walters and Michalski, 2016); by taking the $^{18}O$ values for summertime
precipitation water in the western US (-10‰ to -5‰) (Welker, 2000) and the H$_2$O liquid-to-vapor enrichment factor $\varepsilon_{g\text{-}l}$ of +9‰ at 298 K derived from literature with Eq. (B13) (Michalski et al., 2012), $\delta^{18}O\text{-}OH$ is estimated in the range of -35‰ to -30‰ if we ignore the unknown KIF derived from OH oxidation reaction with the vast majority of atmospheric species. The overall $\delta^{18}O\text{-}HONO$ is modelled using the isotope mass balance model

$\varepsilon_{g-l} = -7.68 + 6.71\left(\frac{10^3}{T}\right) - 1.67\left(\frac{10^6}{T^2}\right) + 0.35(\frac{10^9}{T^3})$          (B13)


*Data availability.* All data are available in the manuscript, the supplementary materials, or data repository (DOI: https://doi.org/10.26300/k056-fs32).


*Supplement.*

*Author contributions.* JD and MH conceived the research. JC, JD and MH designed the research. JC carried out field sampling, laboratory sample analyses, data analyses and figure production, as well as conceived and carried out the isotopic box modeling work. BA, CJ, WW, DB, EJ, JK, HM and EH contribute to field sampling. CB helped with laboratory sample
analyses. JC wrote the paper. All authors contributed to the scientific discussions and preparation of the manuscript.

*Competing interests.* The authors declare that they have no conflict of interest.

*Disclaimer.* Any mention of brand names or manufacturers is for information purposes only and does not constitute an endorsement.



*Acknowledgement.* We thank the entire FIREX-AQ team and WE-CAN team, especially Robert Yokelson and Aerodyne
Research Mobile Lab team. We also thank the United States Forest Service for the field support. We are grateful to Ruby Ho
for laboratory support. We are particularly thankful that Thomas Ryerson and Jeff Peischl generously let us use the NOAA
mobile lab and for their support during the WE-CAN field season. This work was supported by funding from
the National Science Foundation (AGS-1351932 to MH) and the National Oceanic and Atmospheric Administration (AC4
Award NA16OAR4310098 to MH and JD).

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






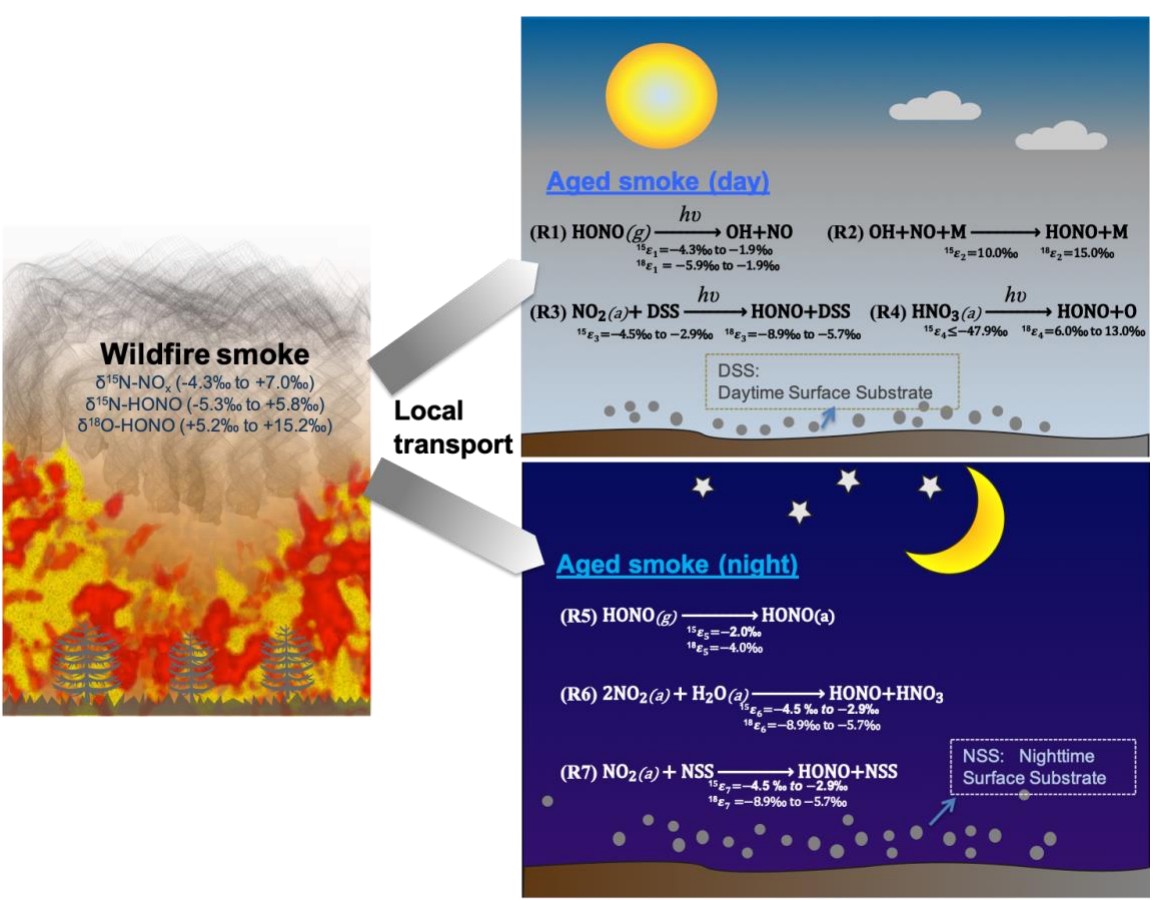

**Fig. 1. The schematics of loss and secondary production of HONO in areas impacted by wildfire smoke in daytime (R1-R4) and nighttime (R5-R7).** We conducted our sample collection <30 km from the edge of the wildfires, with smoke

ages ranging from a few minutes to half a day. M is bath gas including $N_2$, $O_2$, $CO_2$, etc. DSS is daytime substrate surfaces including terrestrial surfaces and aerosol particles that incorporate photoactive metal oxides (e.g. $TiO_2$), humic like organics (e.g. quinone), etc. In essence, solar radiation induces reduction of these substrates with H, and this facilitates H abstraction by $NO_2$ (or H transfer). NSS is nighttime substrate surfaces (terrestrial and aerosol surfaces) containing iron-bearing minerals and/or humic acid (quinone). Note other sinks during both day and night (e.g. OH + HONO) are negligible compared to the

major sinks shown here. Isotopic enrichment factors for N and O result from kinetic isotopic effects associated with each





reaction and are calculated and expressed as $^{15}\varepsilon_i$ and $^{18}\varepsilon_i$, where the subscript number i indicates the reaction number, and the superscripts 15 and 18 denotes the isotopic composition $^{15}N/^{14}N$ and $^{18}O/^{16}O$, respectively.

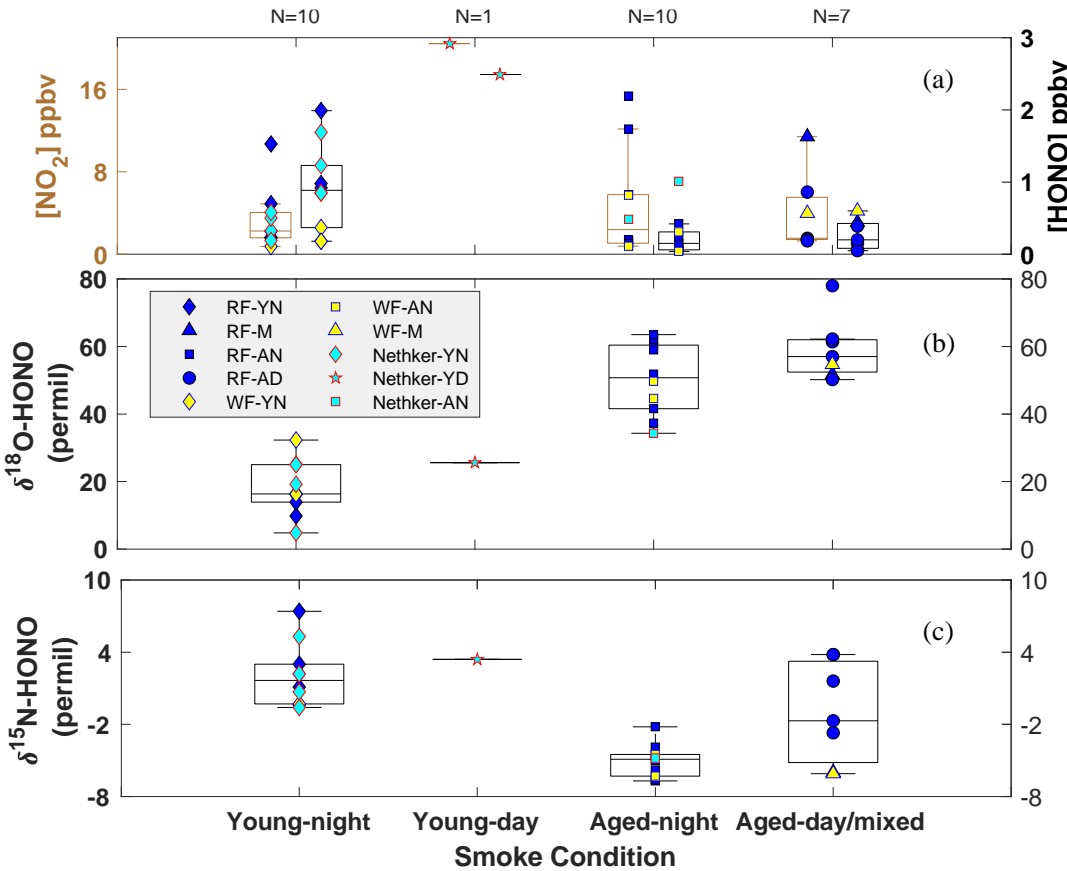

**Fig. 2. Box-whisker plots for concentration of NO₂ (left) and HONO (right) (a), δ¹⁸O-HONO (b), δ¹⁵N-HONO (c) for each sample.** Individual data points are plotted within each box grouped by various field smoke conditions including young nighttime smoke (YN), young daytime smoke (YD), mixed daytime smoke (M) that contains smoke contributed by either night smoke or fresh smoke, aged nighttime smoke (AN), and aged daytime smoke (AD). Data from three wildfires are shown here, including Rabbit Foot (RF) fire during the 2018 WE-CAN campaign, Williams Flats (WF) fire and Nethker fire

during the 2019 FIREX-AQ campaign. N is sample number measured for each condition. Each box-whisker presents the 5th, 25th, 50th, 75th, 95th percentile of sample values in each group.

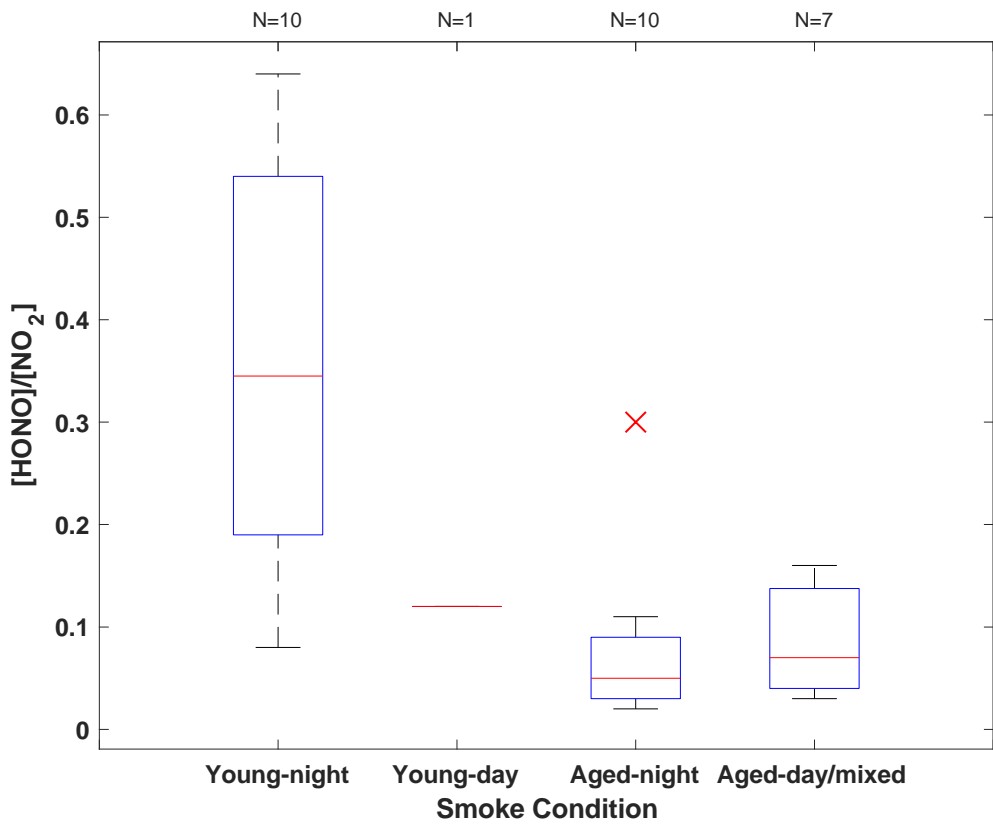

**Fig. 3.** HONO/NO$_2$ concentration ratio summarized in box-whisker plot for each sampling condition. Red cross indicates an

outlier. The whiskers from bottom to top represent 5%, 25%, 50%, 75% and 95% quartiles.

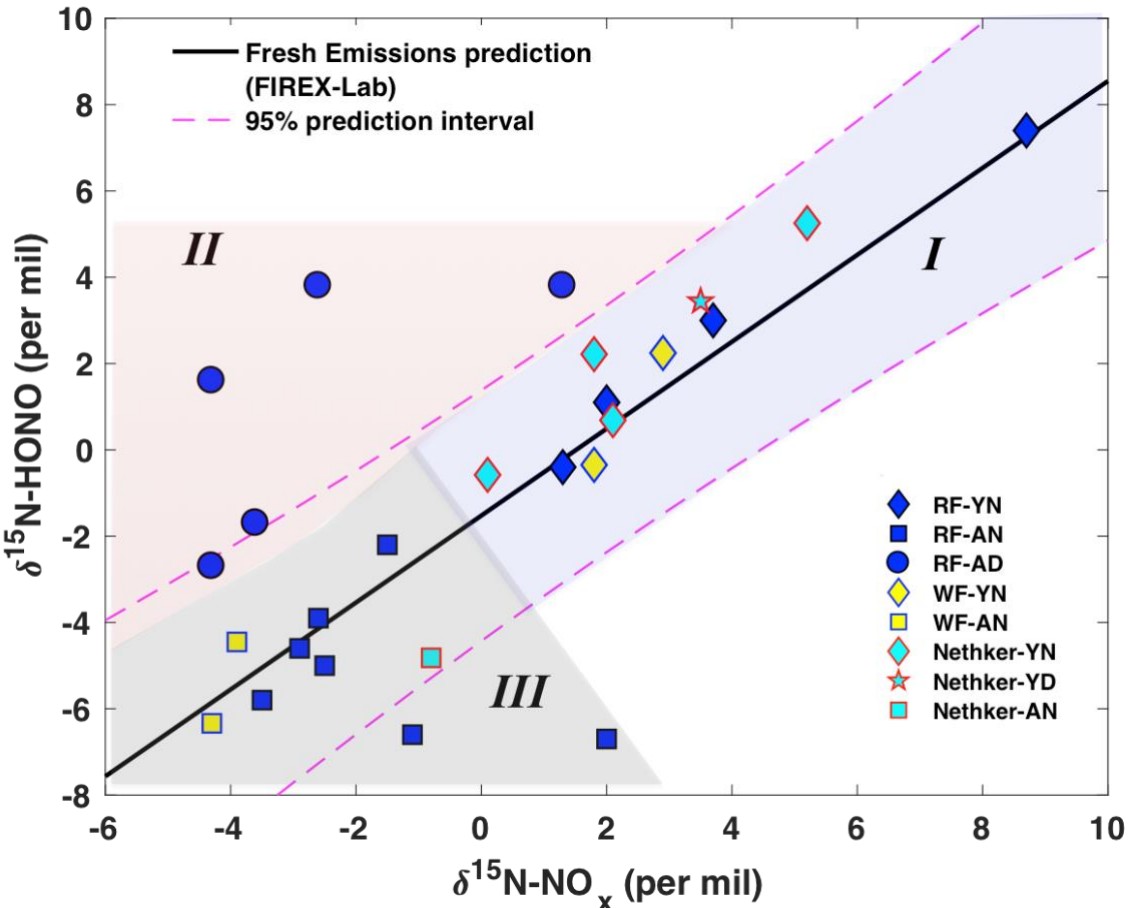

**Fig. 4. Relationship between wildfire-derived $\delta^{15}$N-HONO and $\delta^{15}$N-NO$_x$.** Samples from plumes of three wildfires including Rabbit Foot fire (RF; 2018), Williams Flats fire (WF; 2019) and Nethker fire (2019) are shown as different colors. Different symbols indicate different smoke conditions including young nighttime smoke (YN), young daytime smoke (YD), aged nighttime smoke (AN), and aged daytime smoke (AD). Note that the mixed smoke samples displayed in Fig. 2 are not shown here due to their large uncertainty. The black solid line ($\delta^{15}$N-HONO = 1.01$\delta^{15}$N-NOx − 1.52 (R$^2$ = 0.89, p<0.001) is derived from lab-controlled burning emissions during the 2016 FIREX fire lab study (Chai et al., 2019), and within the 95% confidence interval (magenta dashed lines) predicts much of the field-based $\delta^{15}$N-HONO versus $\delta^{15}$N-NO$_x$. The field data are further grouped into three regimes—young smoke in both day and night (I, light purple shading), aged daytime smoke (II, pink shading), aged nighttime smoke (III, gray shading) based upon the $\delta^{18}$O-HONO results.





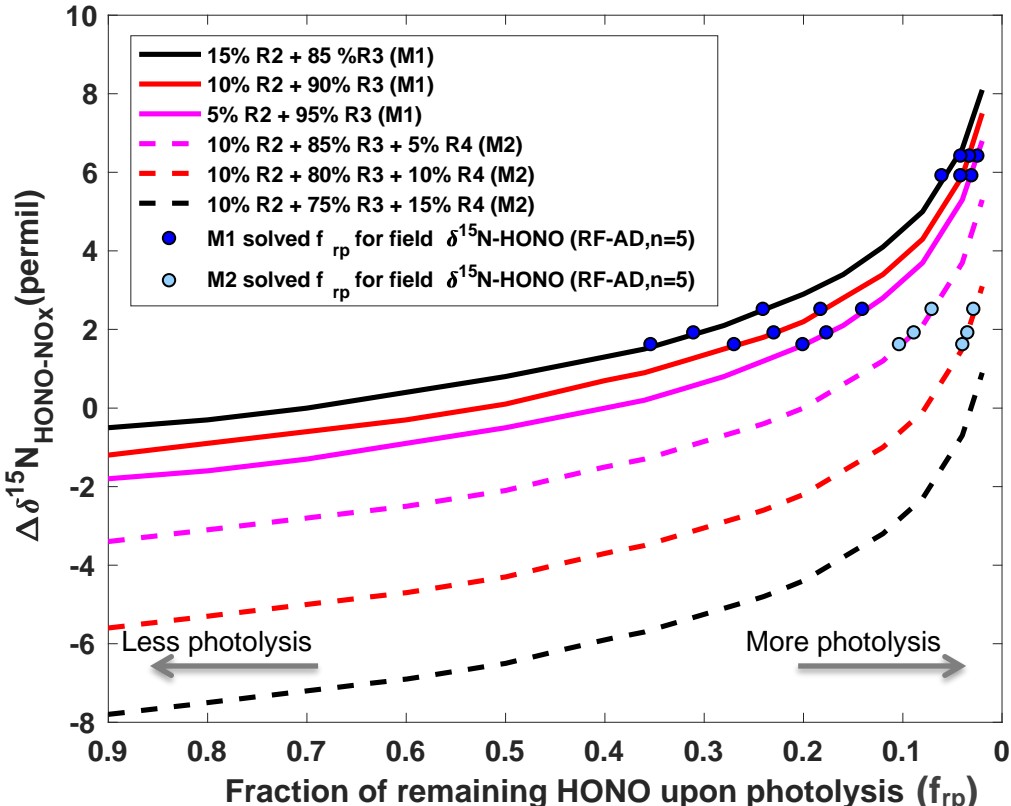

**Fig. 5. Modelling results of $\delta^{15}N$ for aged daytime smoke via two plausible mechanisms (M1 and M2) for secondary**

**HONO production.** The isotope mass balance model (Eq. (1)) is used to simulate the $\delta^{15}N$ difference ($\Delta\delta^{15}N_{HONO-NOx}$

$=\delta^{15}N_{HONO} - \delta^{15}N_{NOx}$) as a function of fraction of HONO remaining after photolysis ($f_{rp}$) in a pseudo-photochemical steady

state. The calculated kinetic fractionation factors used here are explained in Appendix B. In the first mechanism (M1, solid

lines), R3 is the major HONO production pathway with varying relative contribution from R2 (see Fig. 1 for reactions),

which is constrained as producing no more than 15% of the observed HONO concentrations. In the second mechanism (M2,

dashed lines), nitrate photolysis (R4) is included in addition to R2 and R3 for HONO production. Taking the contribution of

R2 of 10% as a constant, three scenarios were modelled by varying the relative contribution of R3 (75%-85%) and R4 (5%-

15%).



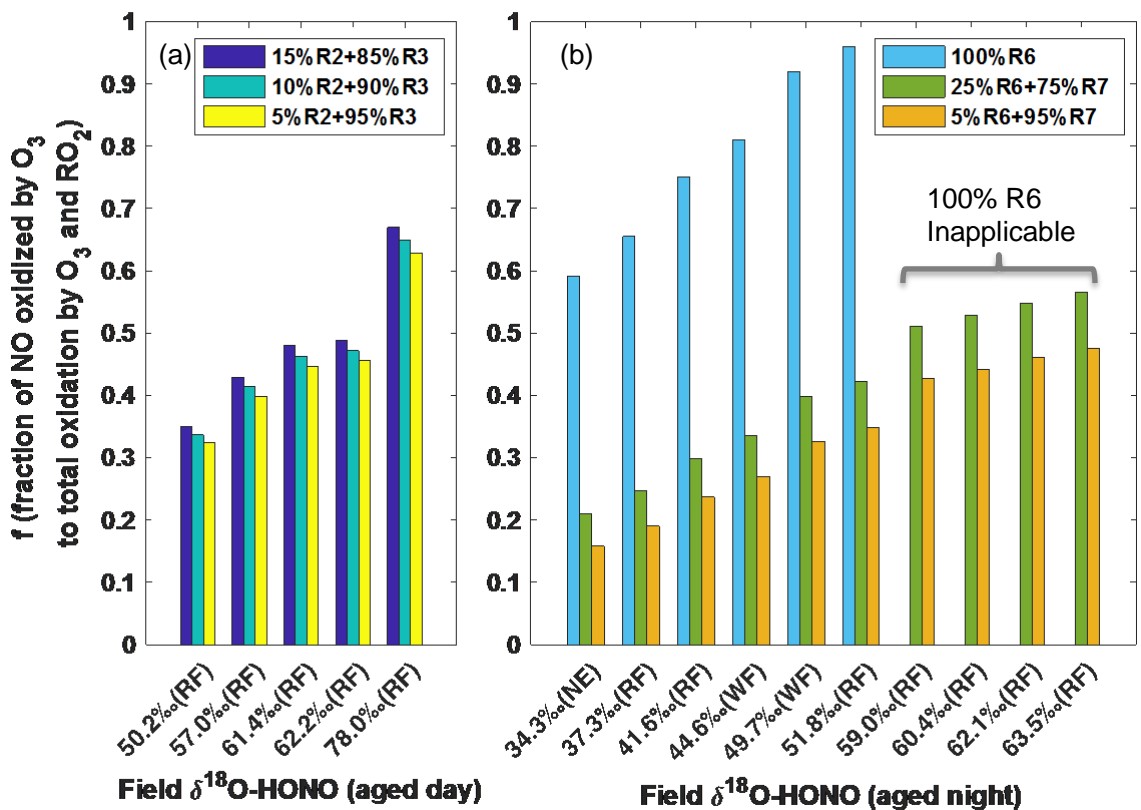

**Fig. 6. Model prediction of fraction of NO oxidized to NO$_2$ via O$_3$ to that via O$_3$ and RO$_2$ together (f$_{O3/(O3+RO2)}^{NO}$) on the**

**basis of field-measured δ$^{18}$O-HONO for aged daytime (a) and nighttime (b) smoke.** During the day (a), the contribution

of R2 to HONO production is varied from 5% to 15% following M1 in Fig. 5, and R3 accounts for the remaining secondary

HONO contribution. The modelling results are shown in Fig. S4 and Table S3 in the supplement. During the night (b), three

scenarios with various contributions of R6 and R7 are modelled (Fig. S5 and Table S4). f$_{O3/(O3+RO2)}^{NO}$ is predicted to be over

unity for the last four observed δ$^{18}$O-HONO values if R6 is assumed as the only nighttime pathway.

**Table 1.** Sampling condition and isotopic composition and concentration results for NO$_x$ and HONO for Rabbit Foot (RF)

fire during the 2018 WE-CAN (a), as well Williams Flats (WF) fire and Nethker fire during the 2019 FIREX-AQ campaign




(b). Smoke conditions include young nighttime smoke (YN), young daytime smoke (YD), mixed daytime smoke (MD), aged nighttime smoke (AN), and aged daytime smoke (AD). The conditions are determined primarily by comparing the field data with the lab data involving three factors: $\delta^{18}O$-HONO, $\delta^{15}N$ relationship between HONO and $NO_x$, HONO/$NO_x$ (or HONO/$NO_2$) ratio, along with the smoke sampling locations. Specifically, significantly elevated $\delta^{18}O$-HONO indicates secondary production of HONO. Note during 2019 campaign, $NO_x$ concentrations were not measured due to instrumental breakdown.

(a)

| Start time (MDT) | End time (MDT) | Fire (smoke condition) | $\delta^{18}O$-HONO | $\delta^{15}N$-HONO | $\delta^{15}N$-$NO_x$ | [HONO] ppbv | [$NO_x$] ppbv | [$NO_2$] ppbv | HONO/$NO_2$ | HONO/$NO_x$ |
|---|---|---|---|---|---|---|---|---|---|---|
| 8/9/18 03:38 | 8/9/18 19:10 | RF (AD) | 50.2‰ | 3.8‰ | 1.3‰ | 0.06 | 1.8 | 1.42 | 0.04 | 0.03 |
| 8/9/18 21:51 | 8/10/18 8:29 | RF (AN) | 37.3‰ | -5.0‰ | -2.5‰ | 0.06 | 1.3 | 1.35 | 0.04 | 0.05 |
| 8/10/18 9:50 | 8/10/18 20:26 | RF (AD) | 61.4‰ | -1.7‰ | -3.6‰ | 0.20 | 1.5 | 1.30 | 0.16 | 0.13 |
| 8/10/18 20:31 | 8/11/18 8:08 | RF (AN) | 60.4‰ | -4.6‰ | -2.9‰ | 0.15 | 1.4 | 1.39 | 0.11 | 0.11 |
| 8/11/18 22:43 | 8/12/18 9:38 | RF (AN) | 51.8‰ | -5.8‰ | -3.5‰ | 0.09 | 1.1 | 1.07 | 0.09 | 0.08 |
| 8/12/18 21:25 | 8/13/18 3:33 | RF (AN) | 62.1‰ | -3.9‰ | -2.6‰ | 0.26 | 0.8 | 0.81 | 0.33 | 0.33 |
| 8/13/18 3:53 | 8/13/18 7:05 | RF (YN) | 16.4‰ | 7.4‰ | 8.7‰ | 0.92 | 1.9 | 1.59 | 0.58 | 0.48 |
| 8/14/18 4:11 | 8/14/18 6:12 | RF (YN) | 16.1‰ | -0.4‰ | 1.3‰ | 0.18 | 1.7 | 1.62 | 0.11 | 0.11 |
| 8/14/18 10:38 | 8/14/18 17:18 | RF (AD) | 57.0‰ | 1.6‰ | -4.3‰ | 0.24 | 1.8 | 1.56 | 0.16 | 0.13 |
| 8/14/18 17:22 | 8/14/18 22:11 | RF (AD) | 78.0‰ | 3.8‰ | -2.6‰ | 0.05 | 1.5 | 1.44 | 0.03 | 0.03 |
| 8/15/18 | 8/15/18 | RF (YN) | 9.8‰ | 1.1‰ | 2.0‰ | 0.98 | 5.5 | 4.90 | 0.20 | 0.18 |





| Start | End | Fire | δ18O | δ15N-HONO | δ15N-NOx | [HONO] | [NO2] | HONO/NO2 |
|---|---|---|---|---|---|---|---|---|
| 8/15/18 0:08 | 8/15/18 4:36 | | | | | | | |
| 8/15/18 5:52 | 8/15/18 7:12 | RF (YN) | 13.9‰ | 3.0‰ | 3.7‰ | 1.99 | 11.7 | 10.70 | 0.19 | 0.17 |
| 8/15/18 19:59 | 8/16/18 9:19 | RF (AN) | 41.6‰ | -2.2‰ | -1.5‰ | 0.15 | 5.9 | 5.78 | 0.03 | 0.03 |
| 8/16/18 15:56 | 8/16/18 17:51 | RF (MD) | 62.2‰ | -2.7‰ | -4.3‰ | 0.39 | 6.5 | 6.03 | 0.07 | 0.06 |
| 8/16/18 21:22 | 8/17/18 6:25 | RF (AN) | 59.0‰ | -6.7‰ | 2.0‰ | 0.42 | 15.6 | 15.34 | 0.03 | 0.03 |
| 8/17/18 8:28 | 8/17/18 10:31 | RF (MD) | 51.7‰ | -6.0‰ | -2.6‰ | 0.44 | 13.5 | 11.40 | 0.04 | 0.03 |
| 8/17/18 21:55 | 8/18/18 9:12 | RF (AN) | 63.5‰ | -6.6‰ | -1.1‰ | 0.25 | 12.3 | 12.15 | 0.02 | 0.02 |


(b)

| Start time (MDT) | End time (MDT) | Fire (smoke condition) | δ18O-HONO | δ15N-HONO | δ15N-NOx | [HONO] ppbv | [NO2] ppbv | HONO/NO2 |
|---|---|---|---|---|---|---|---|---|
| 8/03/19 23:15:57 | 8/04/19 07:27:02 | WF (AN) | 44.6‰ | -4.5‰ | -3.9‰ | 0.31 | 5.7 | 0.05 |
| 8/04/19 18:25:49 | 8/05/19 09:40:08 | WF (AN) | 49.7‰ | -6.3‰ | -4.3‰ | 0.04 | 0.8 | 0.05 |
| 8/06/19 00:20:11 | 8/06/19 09:40:38 | WF (YN) | 16.3‰ | -0.3‰ | 1.8‰ | 0.37 | 0.7 | 0.49 |
| 8/06/19 14:11:24 | 8/06/19 23:02:12 | WF (AD) | 54.7‰ | -6.1‰ | -3.3‰ | 0.60 | 4.0 | 0.15 |
| 8/06/19 23:47:43 | 8/07/19 09:44:16 | WF (YN) | 32.3‰ | 2.2‰ | 2.9‰ | 0.18 | 2.2 | 0.08 |
| 8/09/19 12:32:42 | 8/09/19 14:56:34 | Nethker(YD) | 25.6‰ | 3.4‰ | 3.5‰ | 2.49 | 20.4 | 0.12 |
| 8/10/19 | 8/11/19 | Nethker(YN) | 25.1‰ | 2.2‰ | 1.8‰ | 1.23 | 2.3 | 0.54 |





| | | | | | | | |
|---|---|---|---|---|---|---|---|
| 21:07:49 | 01:47:41 | | | | | | |
| 8/12/19 | 8/12/19 | Nethker(YN) | 25.0‰ | -0.6‰ | 0.1‰ | 1.69 | 3.5 | 0.48 |
| 03:24:15 | 11:24:47 | | | | | | |
| 8/13/19 | 8/14/19 | Nethker(YN) | 4.8‰ | 5.3‰ | 5.2‰ | 0.85 | 4.1 | 0.21 |
| 21:38:03 | 01:28:27 | | | | | | |
| 8/15/19 | 8/15/19 | Nethker (AN) | 34.3‰ | -4.8‰ | -0.8‰ | 1.01 | 3.4 | 0.30 |
| 20:05:55 | 22:43:35 | | | | | | |
| 8/15/19 | 8/16/19 | Nethker(YN) | 19.2‰ | 0.7‰ | 2.1‰ | 0.85 | 1.3 | 0.64 |
| 22:57:07 | 06:28:04 | | | | | | |





**Table A1.** HONO budget estimation.

| Start time (MDT) | Stop time (MDT) | [HONO] ppbv | [NO] ppbv | [NO₂] ppbv | $j_{HONO}$ s⁻¹ | [OH] = 1×10⁶ molecule cm⁻³ | | [OH] = 2×10⁶ molecule cm⁻³ | |
|---|---|---|---|---|---|---|---|---|---|
| | | | | | | $k[OH]$ s⁻¹ | $P_{OH+NO}/L_{HONO}$ | $k[OH]$ s⁻¹ | $P_{OH+NO}/L_{HONO}$ |
| 8/16/18 15:56 | 8/16/18 17:51 | 0.39 | 0.51 | 6.03 | $1.2\times10^{-3}$ | $1.2\times10^{-5}$ | 0.01 | $2.4\times10^{-5}$ | 0.03 |
| 8/9/18 15:38 | 8/9/18 19:10 | 0.06 | 0.41 | 1.42 | $1.1\times10^{-3}$ | $1.2\times10^{-5}$ | 0.08 | $2.4\times10^{-5}$ | 0.15 |
| 8/14/18 10:38 | 8/14/18 17:18 | 0.24 | 0.22 | 1.56 | $1.4\times10^{-3}$ | $1.2\times10^{-5}$ | 0.01 | $2.4\times10^{-5}$ | 0.02 |
| 8/14/18 17:22 | 8/14/18 20:11 | 0.05 | 0.07 | 1.44 | $6.1\times10^{-3}$ | $1.2\times10^{-5}$ | 0.03 | $2.4\times10^{-5}$ | 0.06 |
| 8/10/18 9:50 | 8/10/18 20:26 | 0.20 | 0.22 | 1.30 | $1.1\times10^{-3}$ | $1.2\times10^{-5}$ | 0.01 | $2.4\times10^{-5}$ | 0.02 |

Note: $L_{HONO} = j_{HONO}[HONO]$





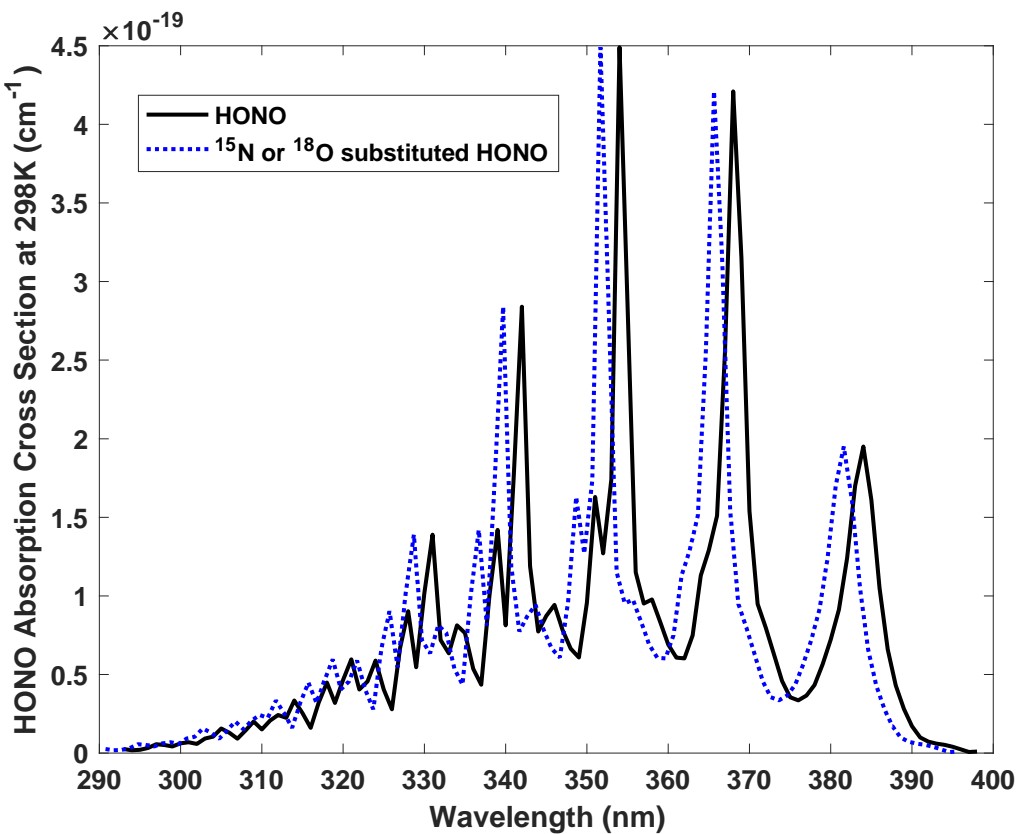

**Figure B1.** Absorption wavelengths shift for HO$^{15}$NO, H$^{18}$ONO and HON$^{18}$O compared with the most abundant form of

HONO (H$^{16}$O$^{14}$N$^{16}$O). The spectra of HO$^{15}$NO, trans-H$^{18}$ONO and trans-HON$^{18}$O are blue shifted 0.23-0.43 nm, 0.21-0.39

nm, and 0.25-0.46 nm respectively spanning 293 to 398 nm. Note that the blue shift illustrated here is 2 nm (larger than the

actual shift) in order to demonstrate the shift clearly.






**Table B1.** Vibrational frequencies of HONO and its isotopologues.

| Vibrational mode | Vibrational frequency of trans-HONO (cm⁻¹) | | | | Vibrational frequency of cis-HONO (cm⁻¹) | | | |
|---|---|---|---|---|---|---|---|---|
| | $\nu$ (t-HONO) | $\Delta\nu$ (t-HO$^{15}$NO) | $\Delta\nu$ (t-H$^{18}$ONO) | $\Delta\nu$ (t-HON$^{18}$O) | $\nu$ (c-HONO) | $\Delta\nu$ (c-HO$^{15}$NO) | $\Delta\nu$ (c-H$^{18}$ONO) | $\Delta\nu$ (c-HON$^{18}$O) |
| v1 (O–H stretch) | 3590.71 | 0.01 | 12.17 | 0.01 | 3426.2 | 0.01 | 11.57 | 0 |
| v2 (O=N stretch) | 1699.76 | 32.5 | 0.27 | 39.28 | 1640.52 | 31.54 | 3.21 | 31.56 |
| v3 (HON bending) | 1263.21 | 1.62 | 10.02 | 1.39 | 1302 | 0.57 | 8.89 | 6.24 |
| v4 (O–N stretch) | 790.12 | 15.73 | 10.76 | 2.14 | 851.94 | 12.47 | 21.38 | 2.57 |
| v5 (O-N-O bending) | 595.6 | 2.88 | 14.91 | 13.96 | 609 | 1.95 | 3.62 | 1.23 |
| v6 (torsion) | 543 | 1.25 | 1.11 | 1.26 | 638.5 | 6.27 | 7.28 | 14.91 |
| $\Delta$ZPE (cm⁻¹) | | 27.00 | 24.62 | 29.02 | | 26.41 | 27.98 | 28.26 |





**Table B2.** Parameters used for TUV solar actinic flux modelling. The modelled enrichment coefficients for HONO

945   photolysis for HO$^{15}$NO, H$^{18}$ONO and HON$^{18}$O are presented in data repository (DOI: https://doi.org/10.26300/k056-fs32).

| MDT | Latitude | Longitude | Altitude (m) | j(HONO) s$^{-1}$ | j(HO$^{15}$NO) s$^{-1}$ | j(H$^{18}$ONO) s$^{-1}$ | j(HON$^{18}$O) s$^{-1}$ | $\varepsilon^{15}$ ‰ | $\varepsilon$ (H$^{18}$ONO) ‰ | $\varepsilon$ (HON$^{18}$O) ‰ |
|---|---|---|---|---|---|---|---|---|---|---|
| 8/16/18 3:56 PM | 44.6726 | -114.2339 | 1700 | 1.412E-03 | 1.408E-03 | 1.408E-03 | 1.407E-03 | -2.6 | -2.2 | -3.0 |
| 8/16/18 5:51 PM | 44.6726 | -114.2339 | 1700 | 9.319E-04 | 9.287E-04 | 9.291E-04 | 9.283E-04 | -3.4 | -3.0 | -3.8 |
| 8/9/18 3:38 PM | 44.5048 | -114.2320 | 1500 | 1.486E-03 | 1.482E-03 | 1.483E-03 | 1.482E-03 | -2.5 | -2.2 | -2.9 |
| 8/9/18 7:10 PM | 45.3870 | -113.9619 | 1117 | 4.733E-04 | 4.724E-04 | 4.726E-04 | 4.722E-04 | -1.9 | -1.4 | -2.3 |
| 8/14/18 10:38 AM | 44.7173 | -114.0226 | 1412 | 1.257E-03 | 1.253E-03 | 1.254E-03 | 1.253E-03 | -2.8 | -2.4 | -3.3 |
| 8/14/18 5:18 PM | 44.7173 | -114.0226 | 1412 | 1.101E-03 | 1.098E-03 | 1.098E-03 | 1.097E-03 | -3.1 | -2.7 | -3.5 |
| 8/14/18 12:00 PM | 44.7173 | -114.0226 | 1412 | 1.496E-03 | 1.492E-03 | 1.492E-03 | 1.491E-03 | -2.5 | -2.2 | -2.9 |
| 8/14/18 1:30 PM | 44.7173 | -114.0226 | 1412 | 1.596E-03 | 1.592E-03 | 1.593E-03 | 1.592E-03 | -2.4 | -2.1 | -2.8 |
| 8/14/18 3:00 PM | 44.7173 | -114.0226 | 1412 | 1.534E-03 | 1.530E-03 | 1.531E-03 | 1.530E-03 | -2.5 | -2.1 | -2.8 |
| 8/14/18 5:22 PM | 44.7173 | -114.0226 | 1412 | 1.085E-03 | 1.082E-03 | 1.082E-03 | 1.081E-03 | -3.2 | -2.7 | -3.4 |
| 8/14/18 8:11 PM | 44.7173 | -114.0226 | 1412 | 1.027E-04 | 1.023E-04 | 1.024E-04 | 1.023E-04 | -4.2 | -3.7 | -4.7 |
| 8/14/18 6:40 PM | 44.7173 | -114.0226 | 1412 | 6.329E-04 | 6.304E-04 | 6.307E-04 | 6.302E-04 | -3.9 | -3.5 | -4.3 |
| 8/10/18 9:50 AM | 45.3870 | -113.9619 | 1117 | 1.052E-03 | 1.049E-03 | 1.046E-03 | 1.045E-03 | -3.2 | -5.5 | -6.3 |
| 8/10/18 8:26 PM | 45.3870 | -113.9619 | 1117 | 7.373E-05 | 7.342E-05 | 7.346E-05 | 7.339E-05 | -4.3 | -3.7 | -4.7 |
| 8/10/18 12:00 PM | 45.3870 | -113.9619 | 1117 | 1.495E-03 | 1.491E-03 | 1.491E-03 | 1.490E-03 | -2.6 | -2.2 | -2.9 |
| 8/10/18 2:00 PM | 45.3870 | -113.9619 | 1117 | 1.592E-03 | 1.588E-03 | 1.589E-03 | 1.587E-03 | -2.4 | -2.0 | -2.8 |
| 8/10/18 4:00 PM | 45.3870 | -113.9619 | 1117 | 1.403E-03 | 1.400E-03 | 1.400E-03 | 1.399E-03 | -2.7 | -2.3 | -3.0 |
| 8/10/18 6:00 PM | 45.3870 | -113.9619 | 1117 | 9.014E-04 | 8.984E-04 | 8.988E-04 | 8.980E-04 | -3.4 | -2.9 | -3.8 |