# Peer review of "Isotopic constraints on wildfire derived HONO"

_Atmospheric Chemistry and Physics, 2021_

## Referee Comment (RC1)

Review of ACP 2021-225, Isotopic Constraints on Wildfire derived HONO, by Chai, et al.,

This paper presents [15]N and [18]O isotopic analyses of HONO and NOx in ground-based samples impacted by wildfire smoke. Unfortunately, it is clear from extensive previous work (Grosjean et al., 1984, Grosjean and Harrison, 1985, and references in Roberts 1990) that the method for sampling NOx has a 1:1 interference from PAN compounds. This renders the NOx measurements reported here invalid since we know that PAN and associated compounds are significant N products in even relatively 'young' wildfire plumes. There might still be information in these measurements that could be a useful addition to our understanding of HONO chemistry in these plumes, but the over-all analysis will need to be completely re-thought. In addition to the above, I found the presentation very difficult to follow, and some sections having to do with HONO alone will need to be extensively revised for the paper to be acceptable. This paper is simply not valid in its current form due to the problem with NOx measurement, and must be rejected for publication at this time. I have the following General Comments that would also need to be addressed, in any future publication, but have refrained from making specific comments.

General Comments

The collection and hydrolysis of PAN in alkaline solution was an early method for the calibration of PAN standards (see for example Stephens, 1969, Grosjean et al., 1984, Grosjean and Harrison, 1985, and references in Roberts 1990) but has somewhat fallen out of favor with the advent of efficient photochemical sources for PANs. The method used for NOx collection in this paper uses the same conditions (e.g. [OH[-]] concentrations) as those PAN collection techniques. Moreover, the alkaline hydrolysis of PANs produces nitrite ion, which will be oxidized by the $KMnO_4$ in the impinger solution in the same manner as the nitrite that arises from NOx collection. Therefore, we can conclude with considerable confidence that PANs will interfere completely with the NOx method used in this work.

One of the biggest issues with the analysis is that the airmass classifications (e.g. YN, YD, AN, AD, and MD) are presented here on the basis of the isotope analysis alone. This imparts a bit of a circular logic to the assignment of these classes. A more complete basis for these classifications apparently was presented in the Kaspari et al 2021 reference, so this should be summarized here for support. Also, if they were initially made using the isotopic analysis, then say so and then support those assertions with other data.

The authors use the notation $f^{NO}_{O3/(O3+RO2)}$, in Equations 4 and 5, but then use $f^{NO}_{O3/RO2}$ in the text – are these meant to be the same thing? If so, this is really confusing. It didn't seem like $f^{NO}_{O3/RO2}$ was defined anywhere else, so I had to assume it was the same as the factor define in Equation 5.

In Figure 1 and associated analysis and discussion around Reactions 6 and 7, the isotope fractionations are the same for both [15]N and [18]O. How then can this support the statements on Lines 361-362 that R6 and R7 lead to very different $\delta^{18}O$ values? – this doesn't make sense on the face of it, and is not at all adequately explained by the material in Appendix B. Is this because of the large difference in $\delta^{18}O$ for $O_3$ relative to $H_2O$? but both reactions 6 and 7 involve $NO_2$ and (which gets an [18]O effect from $O_3$). This whole phenomenon is just not well explained

at all in Appendix B. Also, the nomenclature in Appendix be is faulty, the reactions B10-12, apparently have mixed subscripts that sometimes denote a chemical (e.g. $O_3$ =ozone I assume) and sometimes a reaction (e.g. $O_{6,t}$) and what is 't' in these subscripts?

Nighttime processing of NOx through $NO_3$ and $N_2O_5$ can be quite important chemical pathways to $HNO_3$. Wouldn't these impart an even large $\delta^{18}O$ to the $NO_3^-$ and therefore any HONO derived form that nitrate, since those reactions involve 2 molecules of $O_3$? How would that impact the analysis.

References

Grosjean, D., Fung, K., Collins, J., Harrison, J., and Breitung, E., Portable generator for on-site calibration of peroxyacetyl nitrate analyzers, Anal. Chem., 56, 569-573, 1984.

Grosjean, D., and Harrison, J., Peroxyacetyl nitrate: Comparison of alkaline hydrolysis and chemiluminescence methods., Environ. Sci. Technol., 19, 749-752, 1985.

Roberts, J.M., The atmospheric chemistry of organic nitrates, Atmos. Environ, 24A, 243-287, 1990.

Stephens, E.R. The formation, reactions, and properties of peroxyacyl nitrates (PANs) in photochemical air pollution, Adv. Environ. Sci., 1, 119-146, 1969.

---

## Author Comment (AC1)

We thank Dr. Jim Roberts (reviewer #1) for his time in reading and reviewing our manuscript, and providing references to review. Below are the point-to-point response to the reviewer #1's comments.

1. Discussion on the interference of PAN with NOx collection system

Thanks for raising these concerns. We acknowledge the strong alkaline solution may trap PAN that would be oxidized by in the permanganate solution to nitrate ion. Agreed that at significant concentrations of PAN, i.e., comparable to that of NOx in the atmosphere, and PAN is collected in the permanganate impinger solution it would interfere with the NOx, also collected as nitrate, for isotopic analysis. However, we do not find significant evidence that this is the case in our study conditions.

There is minimum PAN formed in fresh biomass burning (BB) emissions and young smoke of less than half an hour, based upon previous lab and field measurements, as well as modeling studies (Stockwell et al., 2014; Yokelson et al., 2009; Alvarado et al., 2010, 2015). In aged BB plumes in the upper troposphere, PAN can form rapidly at low temperatures and act as a temporary NOx reservoir, reaching a maximum PAN/NOy ratio of 0.3 (comparable to NOx/NOy) within ~2 to 4 hours of aging after emission (Yokelson et al., 2009; Liu et al., 2016; Akagi et al., 2012). For example, Yokelson et al. (2009) measured smoke from Yucatan fires with PAN/NOy varying from 0.11 to 0.3 (average of 0.18) within 2 hours of aging, and similar results were measured by Liu et al. (2016) for agricultural fires in the southeastern U.S. during SEAC4RS, as well as by Akagi et al. (2012). Note these results are all from airborne measurements.

There are no ground-level measurements for PAN in BB plumes during WE-CAN and FIREX-AQ, nor from other field studies, to best of our knowledge. PAN is thermally unstable in the boundary layer during summertime, and its main loss process in the atmosphere is thermal decomposition to release $NO_2$. The lifetime of PAN is on the order of 1 hour or less at 20 °C and above (Talukdar et al., 1995; Fischer et al., 2010). We therefore expected PAN in near-ground air to maintain low levels or less due to photochemistry and thermal decomposition. In addition, our samples integrated over 40 min to 2 hours' time scales, and PAN was less likely to interfere with our NOx results.

Furthermore, although no near-ground PAN measurements in BB plumes are available, the isotopic results can also shed some light on whether PAN interference is important in our case. For aged smoke, we would expect $\delta^{15}$N-NOx to decrease from that in fresh emissions due to partial transformation of NOx to additional oxidized N products (e.g. PAN), as well as isotopic exchange between NOx and these oxidized species; both processes will leave [15]N depletion in NOx and [15]N enrichment in PAN (Walters and Michalski, 2015). If PAN existed at significant concentrations that were 1) comparable with NOx in the atmosphere, and 2) completely collected in the permanganate solution, then the $\delta^{15}$N-$NO_3^-$ would reflect the overall $\delta^{15}$N of NOx + PAN in the final reduced permanganate solution. In this case, we would expect that aged smoke would not shift from the $\delta^{15}$N-NOx range of young smoke, because $\delta^{15}$N shifts in both PAN and NOx could offset each other. However, our observed $\delta^{15}$N-NOx mean values for both aged daytime and nighttime smoke are significantly ($p<0.05$) lower than that of the young smoke (shown in the figure below). This [15]N depletion in NOx indicates the NOx of aged smoke was the predominant N species collected in the permanganate impinger during our field campaign. Similar analysis was also discussed by Miller et al. (2017).

[Figure]

2. "One of the biggest issues with the analysis is that the airmass classifications (e.g. YN, YD, AN, AD, and MD) are presented here on the basis of the isotope analysis alone. This imparts a bit of a circular logic to the assignment of these classes. A more complete basis for these classifications apparently was presented in the Kaspari et al 2021 reference, so this should be summarized here for support. Also, if they were initially made using the isotopic analysis, then say so and then support those assertions with other data."

It is an important challenge in our community as to how to define the age of smoke plumes. We had the unique opportunity there to use the oxygen isotopic composition of HONO ($\delta^{18}$O-HONO) to discriminate young versus old smoke because of the isotopic implications of different oxidants/chemistry. In Lines 92-93, we wrote "In this work, we determined the smoke conditions (young vs aged) primarily by comparing the field $\delta^{18}$O-HONO results with that obtained in our previous lab study that represents fresh emissions,…". In lines 230-243, we presented the approach of using $\delta^{18}$O-HONO to determine young versus aged smoke. For a revised version, we would clarify this approach and include a summary of the analysis in Kaspari et al, 2021, which independently (but still anecdotally) supports our classification based on measurements of other compounds.

3. "The authors use the notation $f^{NO}_{O_3/(O3+RO2)}$, in Equations 4 and 5, but then use $f^{NO}_{O_3/RO_2}$ in the text – are these meant to be the same thing? If so, this is really confusing. It didn't seem like $f^{NO}_{O_3/RO_2}$ was defined anywhere else, so I had to assume it was the same as the factor define in Equation 5."

Sorry about the confusion. These are typos and we will correct all of them to be $f^{NO}_{O_3/(O3+RO2)}$

4. "In Figure 1 and associated analysis and discussion around Reactions 6 and 7, the isotope fractionations are the same for both $^{15}$N and $^{18}$O. How then can this support the statements on Lines 361-362 that R6 and R7 lead to very different d$^{18}$O values? – this doesn't make sense on the face of it, and is not at all adequately explained by the material in Appendix B. Is this because of the large difference in d$^{18}$O for $O_3$ relative to $H_2O$? but both reactions 6 and 7 involve $NO_2$ and (which gets an $^{18}$O effect from $O_3$). This whole phenomenon is just not well explained at all in Appendix B."

Thank you for the question. $\delta^{15}$N of HONO as a product is predominantly determined by the isotopic fractionation shown in Fig. 1. However, for $\delta^{18}$O, we must consider the transfer of isotopic signals upon reaction as well as potential for fractionation of the isotopes. In appendix B.3, lines 590-601 we discuss how to consider both processes and how we determine the transferring isotope effect(s). This will be further clarified in a revised version.

For your specific question on why "R6 and R7 lead to very different d$^{18}$O values", please see lines 593-597, "In R2, OH and NO equally contribute their O-atom to HONO expressed with Eq. (10); In R3 and R7, $NO_2$ is the exclusive O source of HONO while $H_3O^+$ only contribute a H$^+$ to HONO (Ammann et al., 1998; George et al., 2005; Stemmler et al., 2006; Scharko et al., 2017; Kebede et al., 2016); In R6, the hydrolysis mechanism discussed in Appendix B suggests the $H_2O$-derived OH$^-$ and $NO_2$-derived NO$^+$NO$_3^-$ equally contribute their O-atom to HONO (Finlayson-Pitts et al., 2003)."

"Also, the nomenclature in Appendix be is faulty, the reactions B10-12, apparently have mixed subscripts that sometimes denote a chemical (e.g. $O_3$ =ozone I assume) and sometimes a reaction (e.g. $O_{6,t}$) and what is 't' in these subscripts?"

Thank you for noting this. In Lines 590-591, we wrote "For $^{18}O$, in addition to KIF (enrichment factor, $\varepsilon_O^i$ in ‰), $\delta^{18}O$ transferring from different reactants greatly influence $\delta^{18}O$-HONO ($\delta^{18}O_{i,t}$), especially when the two O atoms of HONO are derived from different reactants." 'i' is reaction number, 't' indicates **t**ransfer effect of $\delta^{18}O$. We will add clarification in the text to make this more clear. That said, $\delta^{18}O_{3,t}$ is $\delta^{18}O$ transferring coefficient of R3, where both oxygens in HONO come from $NO_2$.

5.      Nighttime processing of NOx through $NO_3$ and $N_2O_5$ can be quite important chemical pathways to $HNO_3$. Wouldn't these impart an even large $d^{18}O$ to the $NO_3^-$ and therefore any HONO derived form that nitrate, since those reactions involve 2 molecules of $O_3$? How would that impact the analysis.

Thank you for raising this question! Indeed, nighttime processing of NOx to $HNO_3$ is very different from daytime, and this leads to different $\delta^{18}O$ values arising from different isotope transfer effects. When HONO is solely produced from nitrate photolysis, $\delta^{18}O$-HONO would reflect the different $HNO_3$ production pathways. However, when using the best known rates and N isotopic fractionation for this rection and testing different scenarios (i.e., 5%, 10% and 15% of total secondary HONO production), we can only explains the observed $\delta^{15}N$ results with less than 5% of daytime HONO production from nitrate photolysis. Consequently, based upon our $\delta^{15}N$ analysis, we did not incorporate this reaction for $\delta^{18}O$ analysis.

References:

[revised manuscript text omitted]

---

## Author Comment (AC2)

Response to Reviewer #2's comments

The manuscript by Chai et al. reports on ground-based measurements of isotopic ratios ($^{15}N/^{14}N$) and ($^{18}O/^{16}O$) and concentrations of $NO_x$ and HONO derived from fresh and aged wildfire smoke plumes. Measurements were conducted from several locations in the Western U.S. during the WE-CAN and FIREX-AQ field campaigns using state-of-the art measurement techniques. Furthermore, the data is presented and assessed thoroughly to the full extent that the data allows. This is a significant contribution for the following reasons: It reports for the first time the isotopic ratios of HONO in wildfire plumes and the isotopic evidence is used to evaluate the relative importance of various HONO formation/loss pathways (homo- and heterogeneous) that have until now only been studied in the laboratory or invoked with considerable speculation. Thus, I feel this work contributes significantly because it provides in situ insights into which HONO formation and loss processes are important in wildfire smoke plumes. In addition, the authors present a simple but elegant box model for assessing the importance of these pathways, that can be useful in future studies aimed at studying atmospheric processes involving reactive nitrogen. The paper is not without its weaknesses. Most significantly, many of the parameters needed to model (e.g., the enrichment factors) are not well constrained. However, the authors use well-reasoned assumptions and qualify their estimates by clearly discussing the limitations in the extensive appendices to the manuscript. Overall, I feel this is not a deal-breaker since these are the best estimates that can be made using the available data (none of the enrichment factors have been evaluated in the literature). I feel this manuscript should be published in ACP after the following specific points have been addressed.

We really appreciate the careful read, positive feedback and encouragement from Reviewer 2. Below we respond to the reviewer's specific comments point by point in blue text.

The more significant questions in my reading of the work have to do with how HONO is modeled. If I am not mistaken, the isotopic model uses reactions R1-R4 for daytime chemistry and reactions R5-R7 to represent the nighttime chemistry controlling the HONO isotopic signature. In reality, reactions R5-R7 are also occurring during the daytime and could be important. For example, modeling studies often find that good agreement between model and measured HONO concentrations is only possible when deposition processes are included during the daytime (in addition to photolysis). Particle scavenging in smoke events will be particularly important due to the added surface area provided by particulate matter/smoke particles. For the same reason, non-photochemical sources such as R6 will occur during both the night and daytime. I feel it would be useful for the authors to justify their decision to omit reactions R5-R7 in the modeled daytime results. I also wonder how reliable the models results are with respect to distinguishing between Reactions (R6) and (R7)? That is, it was not clear how the parameterization of these two reactions was different and whether, due to the level of uncertainty associated with the enrichment factors and mechanisms, whether it is even possible to distinguish between them, especially since the relative contribution of R6 may be so low. Modern laboratory experiments (and theory) conducted under atmospherically relevant conditions suggest that reaction R6 is only important at very high (>100 ppbV) $NO_2$ concentrations when dimerization is favored. Measured $NO_2$ concentrations in this study were below 20 ppbV, so I would have my doubts that $NO_2$ levels were high enough to favor any $NO_2$ hydrolysis. In addition, in section B.1.2., I agree that HONO desorption involving breaking of the complex $HONO...(H_2O)_n$ is likely important for determining KIF. I note that the distinction between the heterogeneous $NO_2$ reactions (R3, R6, and R7) is the role of water. In R3 & R7, $H_2O$ is the medium, while in R6 $H_2O$ is both reactant and medium, so would one not expect R6 to have a very different enrichment factor compared to R3 and R7?

Thank you for raising these concerns, which are important points to be considered.

First, we agree deposition of HONO could be an important sink during the day. In fact, we have estimated the relative importance of HONO deposition on the ground compared to daytime HONO photolysis. The deposition coefficient ($k_d$) was calculated following equation $k_d = v_{HONO}/H$, where $v_{HONO}$ is the dry deposition velocity and it is assumed to be 0.008 m s$^{-1}$ (Nie et al., 2015), and $H$ is the daytime boundary layer height with a range of 1000-3000 m (Zhang et al., 2020). Taking an average of HONO photolysis coefficient of 0.001 s$^{-1}$, HONO lost to deposition is less than 1% that lost to photolysis. Similarly, HONO lost to OH+HONO and particle uptake is at the same magnitude of deposition. As such in the manuscript we state photolysis is the dominant loss pathway for HONO.

Second, for N isotopic fractionation associated with HONO production, R3, R6 and R7 are not distinguishable because the kinetic processes are all expected to be controlled by a desorption step, as discussed in lines 636-639: "Kinetic isotopic fractionation (KIF) associated with photo-enhanced $NO_2$ conversion is not known. Similar to the nighttime heterogeneous $NO_2$ conversion, R3 is also expected to occur in the surface aqueous phase and the overall KIF is largely determined by that associated with the desorption of HONO from aqueous to gas phase. Thus, $^{15}\varepsilon_3$ and $^{18}\varepsilon_3$ are the same as that of R6 and R7 (Appendix B.1.2)."

From our model and the parameterization for N isotopes, there is not a satisfying way to distinguish R3 and R7 during the daytime. However, we are currently undertaking a series of laboratory studies that aims to characterize if these two reactions can be distinguished via N isotopic fractionation. Thus, we cannot rule out the importance of R7 during the daytime with the current parametrization. In order to address this concern, we have added the text in lines 373-381: "However, there are two limitations to the modeling results. First, as the $^{15}N/^{14}N$ fractionation associated with R3, R6 and R7 are not distinguishable with our current parameterization (Appendix B.1.2 and B.2.2), we cannot rule out the potential importance of heterogeneous $NO_2$-to-HONO conversions (R6 and R7) in daytime. Second, it should be noted that the results represent our best estimate of the average relative importance of R2-R4 for HONO production during our HONO sampling periods (2-10 hours) for the aged daytime plume. Due to the long sample integration time, our samples were influenced by both aged smoke and near-background air when the smoke was very diluted. Under the $NO_x$–limited condition (low $NO_x$ <1 ppbv) in remote background air, nitrate photolysis is expected to be the major secondary HONO source (Ye et al., 2016; Zhou et al., 2011), which cannot be ruled out by our results. Isotopic measurement techniques with higher time resolution will be required to achieve real-time quantification of the HONO budget."

Third, although R6 and R7 cannot be distinguished by N isotopes, the O isotopic signature can be used to distinguish these processes based upon different reaction mechanisms (i.e., oxygen transfer). In lines 326-327, we explained "For δ$^{18}$O-HONO, we also took into account transferring effect of oxygen from different O-containing reactants that produce HONO (as explained in Appendix B)". In lines 334-336, we explained "in addition to kinetic isotopic fractionation, the transferring of δ$^{18}O_{i,t}$ (Eq. (3)) in the reactant (OH, NO, $NO_2$, $H_2O$, and $NO_3^-$) to the product HONO, as HONO contains two O atoms that may stem from more than one reactant (Appendix B)". In lines **409-438** ("The δ$^{18}$O signature is subsequently passed to HONO when it is produced from NO (R2) and $NO_2$ (R3) during the day and from $NO_2$ (R6 and R7) during the night, … and further indicate the important role peroxy radicals play as an oxidant in wildfire smoke impacted environments."), as well as Figure 6, by combining the modeling results and field observations of δ$^{18}$O-HONO in aged nighttime smoke, we showed R7 plays a more important role in $NO_2$-to-HONO conversion. Our result is consistent with the Reviewer's comment that $NO_2$ hydrolysis is less important in the environments where our measurements were conducted.

My last points have to do with readability of the manuscript and figures. The results and discussion refer extensively to reaction equations (R1-R7) and enrichment factors that are only found in boxes within Figure 1. The text chosen for these reactions is a small serif font placed onto a somewhat busy/distracting background; it is very difficult to read and will be even more so in final published form. Because of their importance, I recommend simplifying Figure 1. For example, consider turning it into a (more boring) black-white scheme that omits the graphics and provides all the relevant equations and numbers in an easy-to-read format. I recommend checking references to equations to ensure they are referring to the correct equations. For example, on lines 650-652, there are references to Eqs. (10)-(12); I believe this should be Eqs. (B10)-(B11).

Thank you very much for the suggestions! We added reactions R1-R7 in the text to make the main text more informative and easier for readers to follow. In the text, we also added the references relevant to each of Equations (10)-(12) separately.

*References*
Nie, W., Ding, A. J., Xie, Y. N., Xu, Z., Mao, H., Kerminen, V.-M., Zheng, L. F., Qi, X. M., Huang, X., Yang, X.-Q., Sun, J. N., Herrmann, E., Petäjä, T., Kulmala, M., and Fu, C. B.: Influence of biomass burning plumes on HONO chemistry in eastern China, Atmos Chem Phys, 15, 1147–1159, https://doi.org/10.5194/acp-15-1147-2015, 2015.

Zhang, Y., Sun, K., Gao, Z., Pan, Z., Shook, M. A., and Li, D.: Diurnal Climatology of Planetary Boundary Layer Height Over the Contiguous United States Derived From AMDAR and Reanalysis Data, J. Geophys. Res. Atmospheres, 125, e2020JD032803, https://doi.org/10.1029/2020JD032803, 2020.

---

## Author Comment (AC3)

Response to Reviewer 3

In this manuscript the authors present the ground-based measurement results of concentrations and isotopic ratios ($^{15}N/^{14}N$ and $^{18}O/^{16}O$) of $NO_x$ and HONO in the wildfire smoke plumes in the Western U.S. With a simple box model, they are able to use the data to assess the relative importance of pathways of HONO formation and loss in the smoke plumes. The research approach is innovative and is capable of providing insights into HONO formation mechanisms, although its low method sensitivity limits its applications to air masses with relatively high levels of $NO_x$ and HONO, such as urban atmosphere and wildfire plumes. The paper contains valuable and useful information and thus should be published. I do have some concerns and comments below that need to be addressed before the manuscript is accepted for publication.

We are grateful for the helpful comments from Reviewer 3. Below are the point-to-point response to the reviewer's comments in blue text.

There were simultaneous real-time measurements of HONO, $NO_x$ and other relevant parameters during the study, as stated in the manuscript and published in Kaspari et al. (2021). I suggest the authors to validate the denuder sampling methods by comparing the concentrations of $NO_x$ and HONO with those by Kaspari et al. (2021) and to address the comments by Referee #1 regarding potential interference from PAN on $NO_x$ sampling by denuders. It is critical to prove the methods used to be accurate and reliable before any significant conclusion can be made.

Thank you for the comments and suggestions. Indeed, comparison between the real time measurement and our sample collection is key to ensure accuracy of our offline quantification for both concentration and isotopic composition. During the FIREX fire lab experiment, we applied the same method to quantify the HONO and NOx isotopic composition (Chai et al., 2019). The HONO concentrations captured with our annular denuder system (ADS) were well compared with 4 other high time resolution concentration measurement techniques, including mist chamber/ion chromatography (MC/IC), open-path Fourier transform infrared spectroscopy (OP-FTIR), cavity enhanced spectroscopy (CES), and proton-transfer-reaction time-of-flight mass spectrometer (PTR-ToF). In the same work, the NOx concentration collected in the permanganate impinger was verified by real-time measurement with a chemiluminescence NOx analyzer. In addition, our $NO_x$ collection technique has been verified with real-time NOx concentrations in on-road, near-road and urban background environments (Wojtal et al., 2016; Miller et al., 2017). These agreements verify complete capture of HONO and $NO_x$ associated with biomass burning emissions using our technique, which preserve the isotopic signatures without isotopic fractionation during the sampling process.

Based upon the reviewer's suggestion, we added lines 196-206 and lines 233-236 in the main text and Figure S3 in the supplemental materials.

   *lines 196-206*: "Note that complete collection of HONO and $NO_x$ have been verified in various environments including biomass burning emissions. During the FIREX fire lab experiment, we applied the same method to quantify the HONO and $NO_x$ isotopic composition (Chai et al., 2019). The concentrations of HONO captured with our annular denuder system (ADS) well compared with 4 other high time resolution concentration measurement techniques, including mist chamber/ion chromatography (MC/IC), open-path Fourier transform infrared spectroscopy (OP-FTIR), cavity enhanced spectroscopy (CES), and proton-transfer-reaction time-of-flight mass spectrometer (PTR-ToF). In the same work, the $NO_x$ concentrations collected in the permanganate impinger was verified by real-time measurement with a chemiluminescence $NO_x$ analyzer. In addition, our $NO_x$ collection technique has been verified with real-time $NO_x$ concentrations in on-road, near-road and urban background environments (Wojtal et al., 2016; Miller et al., 2017). These agreements verify complete

capture of HONO and $NO_x$ associated with biomass burning emissions using our techniques, which preserve the isotopic signatures without isotopic fractionation during the sampling process."

*lines 233-236*: "The concentration results for the ADS collected [HONO] agree well with that measured via MC/IC in real-time and averaged over the ADS sampling periods (Fig. S3). The good agreement between these techniques sampling the same plumes near the ground, and previous agreement with other HONO and $NO_x$ observation methods suggest the concentrations are accurate (see also Section 2.3)."

To address reviewer #1's comment on possible PAN interference with NOx, we added text in lines 236-248 and lines 294-304. Please also refer to our response to Reviewer #1's comments.

*lines 236-248*: "It is important to also consider possible interference of peroxyacetyl nitrate (PAN) with $NO_x$ collected in the alkaline permanganate solution for biomass burning conditions (Jaffe and Briggs, 2012). There is minimum PAN formed in fresh biomass burning (BB) emissions and young smoke of less than half an hour, based upon previous lab and field measurements, as well as modeling studies (Stockwell et al., 2014; Yokelson et al., 2009; Alvarado et al., 2010, 2015). In aged BB plumes in the upper troposphere, PAN can form rapidly at low temperatures and act as a temporary $NO_x$ reservoir, reaching a maximum $PAN/NO_y$ ratio of 0.3 (comparable to $NO_x/NO_y$) within ~2 to 4 hours of aging after emission (Yokelson et al., 2009; Liu et al., 2016; Akagi et al., 2012). Though we note that these results are all from airborne measurements. There are no ground-level measurements for PAN in BB plumes during WE-CAN or FIREX-AQ, nor from other field studies, to the best of our knowledge. PAN is thermally unstable in the boundary layer during summertime, and its main loss process in the atmosphere is thermal decomposition to release $NO_2$. The lifetime of PAN is on the order of 1 hour or less at 20 °C and above (Talukdar et al., 1995; Fischer et al., 2010). We therefore expected PAN in near-ground air to maintain low levels or less due to photochemistry and thermal decomposition. Thus, given the short lifetime and the sample integration time of over 40 min to 2 hours' timescale, PAN is unlikely to interfere with our $NO_x$ results."

*lines 294-304*: "We note again that, although no near-ground PAN measurements in BB plumes are available, the isotopic results also suggest that PAN interference is not important to the $\delta^{15}N\text{-}NO_x$ results. For aged smoke, we would expect $\delta^{15}N\text{-}NO_x$ to decrease from that in fresh emissions due to partial transformation of $NO_x$ to additional oxidized N products (e.g., PAN), as well as isotopic exchange between $NO_x$ and these oxidized species; both processes will leave $^{15}N$ depleted in $NO_x$ and $^{15}N$ enriched in PAN (Walters and Michalski, 2015). If PAN existed at significant concentrations that were 1) comparable with $NO_x$ in the atmosphere, and 2) completely collected in the permanganate solution, then the $\delta^{15}N$ would reflect the overall $\delta^{15}N$ of $NO_x$ + PAN in the final reduced permanganate solution. In this case, we would expect that aged smoke would not shift from the $\delta^{15}N\text{-}NOx$ range of young smoke, because $\delta^{15}N$ shifts in both PAN and $NO_x$ could offset each other. However, our observed $\delta^{15}N\text{-}NO_x$ mean values for both aged daytime and nighttime smoke are significantly ($p < 0.05$) lower than that of the young smoke, a good indicator of a lack of PAN interference on the isotopic results (see also Miller et al. (2017))."

The authors reported that nitrate photolysis plays only a minor role (<5%) in HONO formation in daytime aged smoke, while heterogeneous $NO_2$-to-HONO conversion contributes 85-95% to total HONO production, followed by OH+NO (5-15%). This finding is in line with what we would expect from our current understanding in HONO chemistry in the environments with moderately elevated $NO_x$ levels. However, it should be pointed out that HONO can be produced by different mechanisms

in different $NO_x$ concentration regimes. Extensive field and laboratory studies in the past 30 years have shown that the HONO budgets can be well predicted and constrained by the reactions of NO and $NO_2$ in the high-$NO_x$ environments. However, other mechanisms, such as photolysis of surface nitric acid and particulate nitrate, may play an important role in the low-$NO_x$ environments. The real-time measurement data reported by Kaspari et al. (2021) (and also the time-series plot in Figure S3) showed very high concentrations of HONO (up to 6 ppb) and $NO_2$ (over 40 ppb) in bands of smoke plumes, in contract to very lower concentrations in the background air outside the plumes. Due to the long sampling times (2-12 hours for HONO and 0.75 – 2.5 hours for $NO_x$) required for the concentration and isotopic measurements, the "averaged" data may not be representative of wildfire smoke plumes, especially when there were significant dilution by background air in the "aged" plume. Cautions should be taken in interpreting the skewed averaged data.

Thank you for raising this point. We acknowledge previous works' findings that under low NOx conditions, nitrate photolysis is an important source of HONO. As the reviewer pointed out, our sample integration time is much longer than real-time concentration measurements, and our samples may contain both wildfire smoke plumes as well as background air. As such, our results obtained from the combination of modeling and field observation represent the average relative importance of R2 - R4 for HONO production. Techniques for measuring isotopic composition of HONO and $NO_x$ with higher time resolution will be required to characterize the temporally and spatially varied secondary HONO formation mechanism. To clarify this point, we added text in lines 373-381:

"However… it should be noted that the result represents our best estimate of the average relative importance of R2 - R4 for HONO production during our HONO sampling periods (2-10 hours) for the aged daytime plumes. Due to the long sample integration time, we expect our samples were influenced by both aged smoke and near-background air when the smoke was very diluted. Under the $NO_x$–limited condition (low $NO_x$ <1 ppbv) in the remote background air, nitrate photolysis is expected to be the major secondary HONO source (Ye et al., 2016; Zhou et al., 2011), which cannot be ruled out by our results. Isotopic measurement techniques with higher time resolution will be required to achieve real-time quantification of the HONO budget."

The manuscript contains two appendixes and a supplement, and it summarizes the key reactions with isotopic fractionation information in a figure. This unusual presentation style is sometime jumpy and confusing. I suggest that some reorganizations of the manuscript should be made to smooth the flow of data presentation and discussion and to made it easier to read.

Thank you for the suggestions. We reorganized the manuscript by adding reactions R1-R7 in the introduction text, which should make it easier for readers to follow, as suggested by Reviewer 2 as well. In addition, we changed the titles of Appendix A to make it more informative.

To justify the structure of our manuscript a little bit more, the first appendix presents a current state of HONO pathways and budget quantification based upon concentration; and the second appendix presents our parameterization of the N and O isotopic fractionation associated with the major HONO pathways. We put these detailed calculations in the appendices so as to simplify the flow of the main text. Lastly, we have also modified the title of the manuscript to be more detailed to clarify the key findings of the work.

Page 6 line 165: the minimum detection limit of 0.07 mM seems too high. It should be 0.07 μM.

Thank you for catching the typo. We have corrected this in the main text.

Page 14 equations (A1) and (A2): what are R and P in the equations? Is R for the rate of production/loss? From the expression of (A2), P should be the fraction of OH-NO reaction to the total HONO production. All the terms in equations should be defined in the text.

Thank you for the suggestions. We have defined all the terms in the main text (lines 469 and 474 respectively):

"…, where $R_{emission}$, $R_{production}$ and $R_{loss}$ are rate of emission, production and loss respectively."

"…the ratio of R2 to the total HONO production ($P_{OH+NO}$)…"

Page 15 equation (A4): Since the sampling was conducted on the ground stations, ground surface should be considered in S/V; it may be important for the heterogeneous HONO production near the ground, especially during the night.

Thank you for the suggestions. We agree that ground surface is very important during the night for HONO production. Given the large particle loadings, it is hard to quantify the overall S/V. However, we added the discussions on this point in the text in lines 515-516:
"ground surface is also expected to play an important role in nighttime HONO production given our ground sampling location", and added a reference (Tuite et al., 2021)…"

Page 16 equation (B3): Should the equation be as follows?

$1/Y_l = 1/α + 1/Γ_b$

The calculations in lines 393-494 do not make sense.

Thank you for catching the typo in the equation (now line 549). We have corrected the typo in equation (B3). In fact, we calculated the fractionation factor with the correct equation in our original work. Our apologies for the typo!

Figure 5: How do you define the fraction of remaining HONO upon photolysis ($F_{rp}$)? For a daytime aged plume arrived at the site from tens km away, >99% of the original HONO would be photolyzed

within a few hours during the transport. So with <1% of HONO remaining upon photolysis, >15% of R4 contribution may still be possible.

We defined $f_{cp}$ as remaining HONO fraction from secondary production as a result of photolysis, and we modified the sentence in lines 355-358:
"We quantify the remaining HONO fraction from secondary production, $f_{rp}$, to represent HONO that has been produced but not yet photolyzed. Thus, the daytime $\Delta\delta^{15}N_{HONO- NOx}$ for aged smoke was simulated as a function of $f_{rp}$ following a Rayleigh-type isotopic fractionation scheme (Fig. 5)".

As the lifetime of HONO during the day is less than an hour due to photolysis, we expect almost all HONO in the aged smoke were produced from secondary pathways. Thus, we conducted simulations of $\delta^{15}N$ under two sets of mechanisms (M1 and M2) by incorporating the estimated isotopic fractionation factors of all the major formation and loss reactions (R1-R4). By using our field-measured values as constraint on the modeling results, we solve $f_{cp}$ for each daytime aged sample. We found that inclusion of nitrate photolysis (R4) would require very fast HONO photolysis, and this will result in very low $f_{cp}$ ,<0.01, <0.006 and <0.002 for 5% R4, 10% R4 and 15% R4 respectively, in order to reproduce the two highest $\Delta\delta^{15}N_{HONO-NOx}$. This suggests the larger nitrate photolysis contributes to HONO production, the less likely the observed HONO levels (hundreds pptv) can be maintained.

In addition, as the reviewer has pointed out, our sample integration time is much longer than real-time concentration measurements. We responded to this question in the reviewer's second point, and we added text in lines 373-381 to clarify this point:
"However, there are two limitations to the modeling results. First, as the $^{15}N/^{14}N$ fractionation associated with R3, R6 and R7 are not distinguishable with our current parameterization (Appendix B.1.2 and B.2.2), we cannot rule out the potential importance of heterogeneous $NO_2$-to-HONO conversions (R6 and R7) in daytime. Second, it should be noted that the results represent our best estimate of the average relative importance of R2-R4 for HONO production during our HONO sampling periods (2-10 hours) for the aged daytime plume. Due to the long sample integration time, our samples were influenced by both aged smoke and near-background air when the smoke was very diluted. Under the $NO_x$–limited condition (low $NO_x$ <1 ppbv) in remote background air, nitrate photolysis is expected to be the major secondary HONO source (Ye et al., 2016; Zhou et al., 2011), which cannot be ruled out by our results. Isotopic measurement techniques with higher time resolution will be required to achieve real-time quantification of the HONO budget."

References:
Chai, J., Miller, D. J., Scheuer, E., Dibb, J., Selimovic, V., Yokelson, R., Zarzana, K. J., Brown, S. S., Koss, A. R., Warneke, C., and Hastings, M.: Isotopic characterization of nitrogen oxides ($NO_x$), nitrous acid (HONO), and nitrate ($pNO_3^-$) from laboratory biomass burning during FIREX, Atmos. Meas. Tech. 12, 6303–6317, https://doi.org/10.5194/amt-12-6303-2019, 2019.

Miller, D. J., Wojtal, P. K., Clark, S. C., and Hastings, M. G.: Vehicle NOx emission plume isotopic signatures: Spatial variability across the eastern United States, J. Geophys. Res. Atmos., 122, 4698–4717, https://doi.org/10.1002/2016JD025877, 2017.

Tuite, K., Thomas, J. L., Veres, P. R., Roberts, J. M., Stevens, P. S., Griffith, S. M., Dusanter, S., Flynn, J. H., Ahmed, S., Emmons, L., Kim, S.-W., Washenfelder, R., Young, C., Tsai, C., Pikelnaya, O., and Stutz, J.: Quantifying nitrous acid formation mechanisms using measured vertical profiles during the CalNex 2010 campaign and 1D column modeling, J. Geophys. Res. Atmos, 126, e2021JD034689, https://doi.org/10.1029/2021JD034689, 2021.

Wojtal, P. K., Miller, D. J., O'Conner, M., Clark, S. C., and Hastings, M. G.: Automated, High-resolution Mobile Collection System for the Nitrogen Isotopic Analysis of NOx, J. Vis. Exp., e54962, https://doi.org/10.3791/54962, 2016.

Ye, C., Zhou, X., Pu, D., Stutz, J., Festa, J., Spolaor, M., Tsai, C., Cantrell, C., Mauldin, R. L., Campos, T., Weinheimer, A., Hornbrook, R. S., Apel, E. C., Guenther, A., Kaser, L., Yuan, B., Karl, T., Haggerty, J., Hall, S., Ullmann, K., Smith, J. N., Ortega, J., and Knote, C.: Rapid cycling of reactive nitrogen in the marine boundary layer, Nature, 532, 489–491, https://doi.org/10.1038/nature17195, 2016.

Zhou, X., Zhang, N., TerAvest, M., Tang, D., Hou, J., Bertman, S., Alaghmand, M., Shepson, P. B., Carroll, M. A., Griffith, S., Dusanter, S., and Stevens, P. S.: Nitric acid photolysis on forest canopy surface as a source for tropospheric nitrous acid, Nature Geosci, 4, 440–443, https://doi.org/10.1038/ngeo1164, 2011.